# An OMA1 redox site controls mitochondrial homeostasis, sarcoma growth, and immunogenicity

Richard Miallot[1], Virginie Millet[1], Yann Groult[1], Angelika Modelska[1], Lydie Crescence[2], Sandrine Roulland[1], Sandrine Henri[1] , Bernard Malissen[1,3], Nicolas Brouilly[4], Laurence Panicot-Dubois[2] , Renaud Vincentelli[5], Gerlind Sulzenbacher[5], Pascal Finetti[6], Aurélie Dutour[7], Jean-Yves Blay[7,8], François Bertucci[6], Franck Galland[1], Philippe Naquet[1] 

**Aggressive tumors often display mitochondrial dysfunction. Upon oxidative stress, mitochondria undergo fission through OMA1-mediated cleavage of the fusion effector OPA1. In yeast, a redox-sensing switch participates in OMA1 activation. 3D modeling of OMA1 comforted the notion that cysteine 403 might participate in a similar sensor in mammalian cells. Using prime editing, we developed a mouse sarcoma cell line in which OMA1 cysteine 403 was mutated in alanine. Mutant cells showed impaired mitochondrial responses to stress including ATP production, reduced fission, resistance to apoptosis, and enhanced mitochondrial DNA release. This mutation prevented tumor development in immunocompetent, but not nude or cDC1 dendritic cell–deficient, mice. These cells prime CD8$^+$ lymphocytes that accumulate in mutant tumors, whereas their depletion delays tumor control. Thus, OMA1 inactivation increased the development of anti-tumor immunity. Patients with complex genomic soft tissue sarcoma showed variations in the level of OMA1 and OPA1 transcripts. High expression of OPA1 in primary tumors was associated with shorter metastasis-free survival after surgery, and low expression of OPA1, with anti-tumor immune signatures. Targeting OMA1 activity may enhance sarcoma immunogenicity.**

## Introduction

Mitochondria operate as major hubs regulating cell life and death. By providing energy through the electron transport chain (ETC), they also generate reactive oxygen species (ROS) that may be detrimental to tumor survival. Functional and/or structural alterations in these organelles frequently occur in cancer and witness the occurrence of mitochondrial stress (McBride & Soubannier, 2010; Baker et al, 2014). These changes contribute to the metabolic rewiring of cancer cells toward a glycolytic phenotype, a process often driven by oncogenic alterations. Nevertheless, the preservation of mitochondrial activity is required for the metabolic plasticity of tumor cells. It contributes to NAD$^+$ regeneration and to the processing of alternative carbon sources such as glutamine for anabolic pathways (Porporato, 2018).

To optimize their functionality, mitochondria undergo fusion and fission processes, allowing cells to cope with metabolic adaptations and to scavenge damaged organelles. OPA1 is necessary for inner membrane fusion (Alavi, 2019). This protein is highly expressed in metabolically demanding tissues (Gilkerson, 2018). The abundance of the L-OPA1 isoform is down-regulated by proteolytic cleavage at the S2 or S1 sites generating distinct S-OPA1 isoforms by either the constitutively expressed YMEL1 (Song et al, 2007) or the stress-regulated OMA-1 proteases, respectively (Ehses et al, 2009; Head et al, 2009). An additional YMEL1 cleavage site called S3 is involved in the production of another S-OPA1 isoform that contributes to mitochondrial elongation (Wang et al, 2021). Furthermore, the S2 site is required for OXPHOS-induced hyperfusion. The equilibrium between L-OPA1 and S-OPA1 determines the inner membrane fusion potential (Ishihara et al, 2006), whereas S-OPA1 alone contributes to the maintenance of mitochondrial DNA, respiratory complexes, and crista structure (Del Dotto et al, 2017). Upon mitochondrial depolarization or oxidative stress, the activation of OMA1 leads to the cleavage of several target proteins including OPA1 at the S1 site (Baker et al, 2014; Murata et al, 2020) and DELE1 (Fessler et al, 2020; Guo et al, 2020), provoking mitochondrial fission and integrated stress response, respectively. Then, the peripheral mitochondrial division is triggered (Kleele et al, 2021) and leads to the clearance of damaged mitochondria by mitophagy (Twig & Shirihai, 2011). Furthermore, OMA1 links mitochondrial protein

[1]Aix-Marseille Université, INSERM, CNRS, Centre d'Immunologie de Marseille-Luminy, Marseille, France  [2]Aix Marseille Université, INSERM 1263, INRAE 1260, Plateforme d'Imagerie Vasculaire et de Microscopie Intravitale, C2VN, Marseille, France  [3]Centre d'Immunophénomique, Aix Marseille Université, INSERM, CNRS, Marseille, France  [4]Aix-Marseille Université, CNRS, IBDM, Marseille, France  [5]Aix-Marseille Université, CNRS, Architecture et Fonction des Macromolécules Biologiques, Marseille, France  [6]Laboratory of Predictive Oncology, Centre de Recherche en Cancérologie de Marseille (CRCM), Institut Paoli-Calmettes, Aix-Marseille Université, Centre National de la Recherche Scientifique (CNRS), Institut National de la Santé et de la Recherche Médicale (INSERM), Marseille, France  [7]Childhood Cancers and Cell Death Laboratory, Cancer Research Center of Lyon (CRCL), INSERM 1052, CNRS, Lyon, France  [8]Department of Medicine, Centre Léon Bérard, UNICANCER & University Lyon I, Lyon, France

Correspondence: naquet@ciml.univ-mrs.fr; miallot@ciml.univ-mrs.fr

quality control to retrograde signaling that is required for the induction of tolerance mechanisms to stress (Bohovych et al, 2016; O'Malley et al, 2020).

OMA1 maturation is a tightly regulated process allowing its integration in the inner mitochondrial membrane; both proteolytic and autocatalytic processes lead to the production of enzymatically active OMA1 protein isoforms (Baker et al, 2014; Zhang et al, 2014; Consolato et al, 2018). Interestingly, distinct regions of the OMA1 protein are involved in stress sensing. A recent report in yeast identified a redox-sensing site required for the production of active OMA1 upon mitochondrial depolarization and oxidative stress (Bohovych et al, 2019). This regulation depends on the formation of a disulfide bridge between cysteines 272 and 332 of yeast OMA1 and contributes to the organization and function of the ETC. In this model, the loss of OMA1 function results in increased ROS production and impaired retrograde signaling required for cell survival. We reasoned that by interrupting OMA1 function in cancer cells, one might interfere with the induction of stress response pathways and enhance cancer cell death. Because the above-mentioned Cys are conserved in the mammalian *oma1* gene, we decided to evaluate their contribution to the regulation of OMA1 activity and mitochondrial function in sarcoma.

*Oma1* is a candidate modulator of cancer progression. Its overexpression is of poor prognosis in gastric carcinoma (Amini et al, 2020), and breast and squamous cell lung carcinoma (Alavi, 2019), but conversely associated with improved survival in lung adenocarcinoma (Alavi, 2019) and breast carcinoma (Daverey et al, 2019). In colorectal cancer, OMA1 supports metabolic reprogramming under hypoxic conditions (Wu et al, 2021). These results indicate that OMA1 involvement in cancer depends upon the context, suggesting a differential involvement of mitochondrial activity in these tumors. We previously showed that preservation of mitochondrial fitness limited mouse fibrosarcoma progression (Giessner et al, 2018). In sarcoma patients, the contribution of mitochondrial activity to tumor progression is still debated, an issue complicated by the heterogeneity of these tumors (Miallot et al, 2021). We therefore probed the involvement of the OMA1/OPA1 pathway in a mouse sarcoma model and tested whether the level of *OMA1* and *OPA1* expressions was associated with clinical outcome and immune variables in human soft tissue sarcomas (STS).

## Results

### The C403A mutation abrogates OMA1-dependent OPA1 cleavage in a mouse sarcoma model

OMA1 processing is recapitulated in Fig 1A and based on a 3D model of the mouse 59-kD pre-pro-OMA1 protein available in the AlphaFold database (https://alphafold.ebi.ac.uk/). This immature form undergoes further proteolysis during mitochondrial insertion (see Fig 1 legend). Upon depolarization induced by the carbonyl cyanide m-chlorophenyl hydrazine (CCCP) uncoupling drug, the mature 40-kD L-OMA1 isoform is autocatalytically cleaved at the C-terminal end to generate the 35-kD S-OMA1. This isoform is catalytically active on its target proteins, undergoes further C-terminus

cleavage, and is unstable (Baker et al, 2014; Zhang et al, 2014; Alavi, 2021). As shown in yeast, this activation process might involve redox changes in cysteines 272 and 332, corresponding to cysteines 403 and 461 in mouse OMA1 (Bohovych et al, 2019). Because mutation of yeast Cys332, equivalent to mouse Cys461, provoked a loss of OMA1 stability, we focused our efforts on Cys403. We addressed the contribution of these cysteines to OMA1 activation using biochemical and molecular approaches. We first produced a recombinant OMA1 protein displaying only the outer membrane domain that contains the catalytic site, coupled to the protein disulfide-isomerase DsbC to optimize the production in bacteria (Fig S1A). The recombinant OMA1 protein cleaved an artificial substrate based on a short OPA1 peptide containing the OMA1-specific cleavage site (Tobacyk et al, 2019). However, the specificity of this cleavage has been questioned (Alavi, 2022). Accordingly, the activity is only partially inhibited by the N,N,N',N'-tetrakis(2-pyridylmethyl)ethylenediamine (TPEN), a zinc chelator used as a negative control of the Zn-dependent OMA1 activity (Fig S1B). Unfortunately, the production of a protein in which Cys403 was replaced by an Ala (C403A mutant) did not lead to the production of a stable protein (Fig S1A). Because no experimental 3D structure of OMA1 is available, we took advantage of recent developments in artificial intelligence (AI) and inspected the 3D model of mouse OMA1 generated by the AlphaFold algorithm (Jumper et al, 2021), which revealed that the disulfide bridge in question is located far from the catalytic site and exposed at the protein surface (Fig 1A). It is worth mentioning that significant portions of the protein, mostly external loops, are modeled with a very low confidence level, suggesting structural disorder and plasticity. We then generated a model of the OMA1 C403A mutant using DeepMind's Colab notebook (Jumper et al, 2021). Comparison of AlphaFold models between WT and mutated OMA1 revealed that no major structural changes were induced by the C403A mutation. It should, however, be noted that the model of the OMA1 C403A mutant generated by DeepMind's Colab notebook might be biased toward the training provided by the model of native OMA1, not accounting for subtle modifications engendering major structural changes in vivo (Fig S1C).

We then mutated Cys403 into Ala in the murine fibrosarcoma cell line MCA205 by prime editing (Anzalone et al, 2019). We obtained heterozygote and homozygote 403C>A mutant clones (named C403A throughout the study) and control unedited cells (named CTRL throughout the study; Fig S1D and E for DNA sequencing profile). We performed several control experiments to validate the edition process. First, as expected, *Oma1* and *Opa1* transcript levels were unchanged in edited cell lines (Fig S1F). Furthermore, all edited lines had a comparable although not equivalent growth rate in vitro (Fig S1G). Although the experiments were performed with several independent clones (Fig S1G), we chose representative CTRL and C403A clones to illustrate our results.

We then evaluated the expression of OMA1 protein isoforms in control or CCCP-treated cells. In MCA205 sarcoma cells at a steady state, both the immature and the active S-OMA1 isoforms are detectable in total cell extracts (Fig 1B and C). In mitochondrial extracts, only L- and S-OMA1 were detected, with a predominance of the S-OMA1 isoform (Fig 1D) that may witness the presence of an endogenous mitochondrial stress associated with OMA1 autocatalysis in this tumor cell line. In C403A unstimulated total cell and

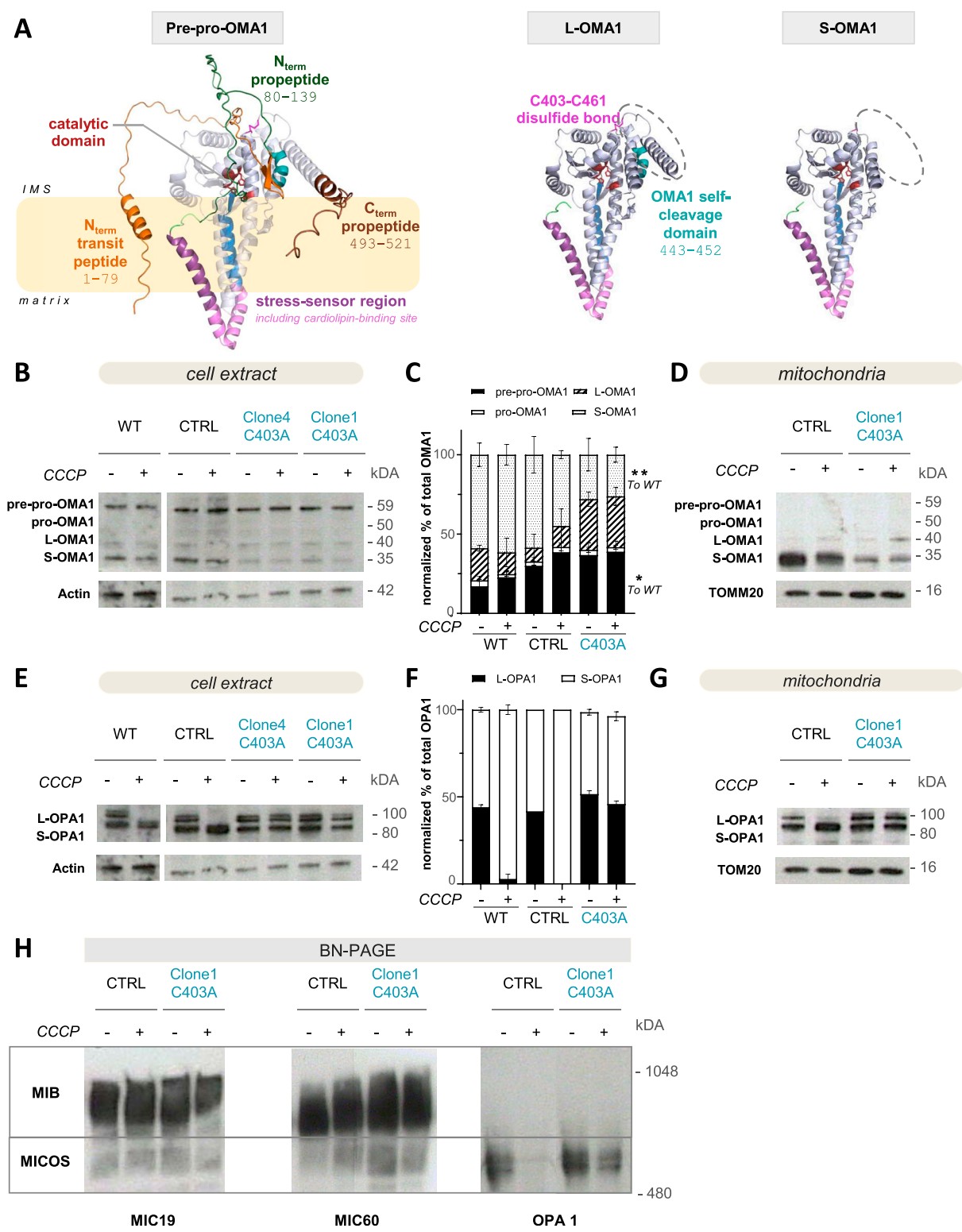

**Figure 1. C403A mutation alters OMA1 maturation and catalytic activity.**

**(A)** OMA1 model, retrieved from the AlphaFold tool, and representation of the maturation processing of OMA1. OMA1 is a mitochondrial protein with a M48 metalloendopeptidase domain facing the mitochondrial intermembrane space. It probably exists as a homo-oligomeric complex, and the activation mechanism remains incompletely understood (Levytskyy et al, 2017; Alavi, 2021). The cysteine residues of interest are highlighted in pink. OMA1 maturation follows the following steps: the transit peptide in orange, facing the mitochondrial matrix from pre-pro-OMA1, is cleaved between 79L and 80S residues to obtain the pro-OMA1 form. The propeptide in dark green is then cleaved between residues 139Q and 140A and in the C-terminal region after residue 493, in brown. This process requires the AFG3L2 and YMEL1 proteases (Rainbolt et al, 2016; Consolato et al, 2018). The mature long form of OMA1 (L-OMA1) is autocatalytically cleaved into the S-OMA1 form through the cleavage of a

mitochondrial extracts, the relative proportion of S-OMA1 was significantly reduced compared with CTRL extracts (Fig 1B–D). Because OMA1 is subjected to autocatalysis, we tested whether this activity was preserved in C403A OMA1. We performed a kinetic analysis of OMA1 processing after CCCP exposure in both cell types (Fig S1H). In CTRL mitochondrial extracts, the L-OMA1 isoform progressively disappeared and 35- and 33-kD S-OMA1 isoforms progressively appeared after 1 h of treatment in agreement with the autocatalytic activity. In C403A cells, the 35-kD S-OMA1 isoform was present in reduced amounts, and at late time points, the shorter OMA1 isoforms were not detectable. This might be due to the fact that C403A cells show more signs of damage after prolonged CCCP treatment than CTRL cells. These results indicate that the C403A mutation enhances S-OMA1 instability and might affect its autocatalytic activity.

We then investigated whether the C403A mutation prevented the cleavage of the OMA1–target protein OPA1. At the steady state, the OPA1 long isoform (L-OPA1) is cleaved by YMEL1 at the S2 or S3 site into a short 80- to 90-kD isoform (S-OPA1), thereby contributing to the regulation of the homeostatic fusion/fission process (Song et al, 2007; Head et al, 2009; Wang et al, 2021). This explains the presence of comparable levels of L- and S-OPA1 isoforms under basal culture conditions of the cell line (Fig 1E). Activated S-OMA1 cleaves L-OPA1 at the S1 site to generate another short inactive S-OPA1 isoform (Song et al, 2007). We evaluated the proportion of S-OPA1 and L-OPA1 as an indicator of CCCP-induced OMA1 activation in total cell extracts. Under CCCP-induced stress, OPA1 was fully converted into S-OPA1 (Fig 1E and F). In contrast, untreated or CCCP-treated C403A cells displayed a constant proportion of S- and L-OPA1 isoforms. These results were confirmed using mitochondrial extracts (Fig 1G). Therefore, C403A OMA1 is unable to cleave its mitochondrial OPA1 substrate upon stress. Unexpectedly, extracts from both CTRL and C403A cells could cleave the reporter peptide with equivalent efficacy, and this cleavage was inhibited by TPEN (Fig S1C) (Tobacyk & MacMillan-Crow, 2021). Keeping in mind the poor specificity associated with the reporter peptide, this finding suggested that modifications induced by the C403 mutation might be preventing the engagement of OMA1 with its target proteins.

The OMA1 and OPA1 proteins are part of the mitochondrial contact site and crista organizing system (MICOS) complex that regulates crista structure (Glytsou et al, 2016; Huynen et al, 2016; Viana et al, 2021). The lack of OMA1 was shown to reduce the stability of the MICOS complex (Viana et al, 2021). BN-PAGE analysis revealed no significant changes between control and C403A cells in the proportion of the mitochondrial bridging (MIB) complex and MICOS supramolecular complexes under basal or CCCP-treated conditions, showing that the C403A mutation does not prevent their formation (Figs 1H and S1I). OPA1 could be detected in the MICOS complex, and interestingly, its proportion was reduced after CCCP

treatment of control but not C403A cells, as expected because of the lack of OMA1-mediated cleavage.

## Loss of mitochondrial homeostasis in stressed OMA1 mutant cells

We evaluated the impact of the C403A OMA1 mutation on mitochondrial organization. We performed a high-resolution confocal analysis of MCA205 cells stained with MitoTracker Deep Red and reconstructed the mitochondrial network in 3D. Under basal conditions, both CTRL and C403A OMA1 cells showed a complex mix of small individual mitochondria and tubular mitochondrial networks (Fig 2A). After CCCP-induced mitochondrial membrane depolarization, CTRL cells showed an increased mitochondrial fission index in agreement with OPA1 inactivation. In contrast, the mitochondrial network and index remained unchanged in CCCP-treated C403A cells (Figs 2A and B and S2A for data on additional edited clones). Because MitoTracker Deep Red depends on transmembrane potential for uptake (Xiao et al, 2016), we confirmed our results using translocase of the outer mitochondrial membrane 20 (TOMM20) staining on fixed and permeabilized cells (Nakashima-Kamimura et al, 2005). As shown in Fig S2B, C403A cells showed a network of hypertubular mitochondria that were unaffected by CCCP treatment unlike that of CTRL cells. This result confirmed the loss of stress-induced adaptation of mitochondrial reorganization.

Because OMA1-dependent regulation is required for the maintenance of mitochondrial fitness, we scored several parameters of mitochondrial function in vitro. Despite higher basal ECAR and OCR in C430A versus CTRL cells (Fig S2C), the ATP rate index reflecting mitochondrial versus glycolytic-dependent ATP production was reduced in C403A compared with CTRL cells (Fig 2C and S2D). Overall, this indicated that whereas the lack of mitochondrial fragmentation could be associated with a relative increase in OXPHOS (Wu et al, 2021), C403A cells rather relied on glycolysis to sustain their hypermetabolic profile. Quantification of mitochondrial ROS overproduction after inhibition of complex III of the ETC complex by antimycin A showed that C403A cells had a reduced leakage of mitochondrial ROS compared with CTRL cells (Fig 2D). We then quantified mitochondrial depolarization and mass using the MitoTracker Deep Red (MDR) and MitoTracker Green (MG) probes, respectively (Xiao et al, 2016). As shown in Fig S2E, CTRL and C403A clones showed comparable levels of MDR and MG staining in culture. In vitro CCCP treatment provoked a major reduction in mitochondrial mass and polarization in both cell types. Interestingly, the MDR/MG ratio tended to increase in CTRL but to decrease in C403A cells (Fig 2E), suggesting that mitochondria from C403A cells might be more susceptible to CCCP-induced depolarization.

OMA1-deficient cells were previously shown to be resistant to apoptosis because of the reduced leakage of cytochrome c (Jiang et al, 2014; Gilkerson et al, 2021). We treated CTRL and C403A cells

---

second peptide approximately in the region 443–452, in turquoise (Baker et al, 2014; Zhang et al, 2014). **(B, C, D, E, F, G)** Western blot analysis of OMA1 (B, C, D) and OPA1 (E, F, G) proteins prepared from WT, unedited, and C403A total MCA205 cell (panels B and E) or mitochondrial extracts (panels D and G) exposed for 1 h or not to the uncoupling agent CCCP. Quantification included data obtained from analysis of two WT and four C403A samples (source data), as shown in panels C for OMA1 and F for OPA1 using actin or TOMM20 as control cell or mitochondrial protein (n = 2). Mann–Whitney test; * *P* < 0.05. **(H)** BN-PAGE analysis of native proteins prepared from CTRL and C403A cell mitochondrial extracts. MIB and MICOS complex composition was analyzed using anti-MIC60, anti-MIC19, and anti-OPA1 antibodies (n = 2). Source data are available for this figure.

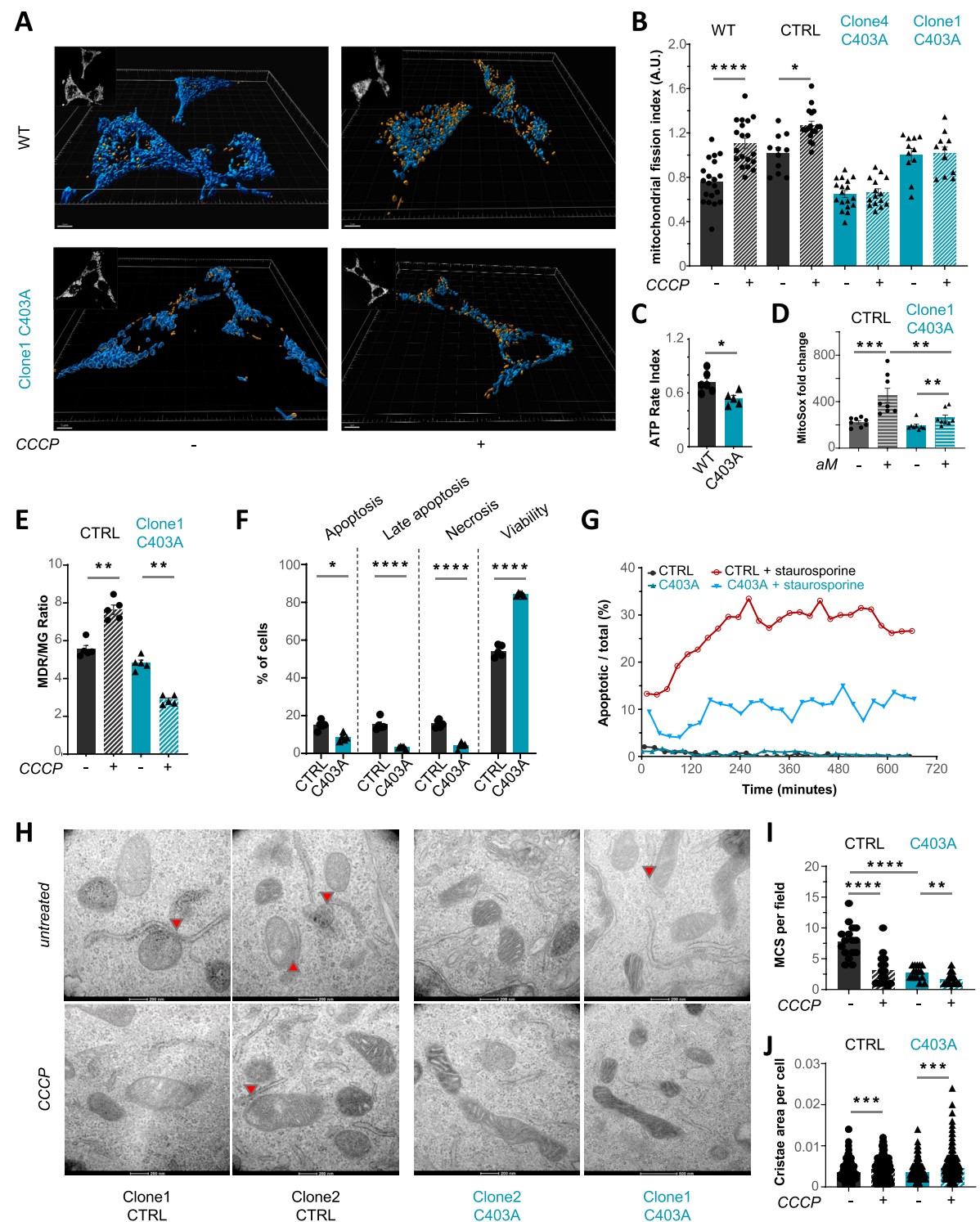

**Figure 2. Evaluation of mitochondrial fitness.**
**(A)** Untreated or CCCP-treated CTRL and C403A MCA205 cells were labeled with mitochondrial depolarization (MDR) (5 μM), and the mitochondrial network was reconstructed by 3D modeling as described in the Materials and Methods section. Yellow and blue dots represent individual mitochondria and network, respectively. Scale bar: 5 μm (n = 2). **(B)** Mitochondrial fission index was calculated in WT, CTRL, and C403A cells in the presence or the absence of CCCP stimulation (n = 2). Two-way ANOVA Tukey's multiple comparisons test was performed for statistical analysis (****P < 0.0001 and * P < 0.05). **(C)** ATP rate index of WT and C403A MCA205 cells was assessed using Seahorse XFp (n = 2). Mann–Whitney test; * P < 0.05. **(D)** Mitochondrial ROS were evaluated by flow cytometry. Data were represented as the mean fluorescent intensity of the MitoSOX probe on CTRL and C403A clones. Mann–Whitney test; ***P < 0.001 and **P < 0.01. **(E)** Flow cytometry quantification of MDR and mass (MG) from untreated or CCCP-treated CTRL and C403A cells. The MDR/MG ratio was calculated (n = 5). Mann–Whitney test; **P < 0.01. **(F)** Flow cytometry evaluation using Annexin V and SYTOX Blue staining of staurosporine-induced CTRL and C403A cell death at 24 h. **(G)** Quantification of apoptosis by holotomographic microscopy of CTRL

with the apoptosis-inducing agent staurosporine and quantified by flow cytometry the fraction of apoptotic cells 24 h later. As shown in Fig 2F, C403A cells were more resistant to apoptosis than CTRL cells. This result was confirmed by live-imaging holotomographic microscopy recording over the first 12 h post-CCCP treatment (Fig 2G). Similar results on apoptosis induction were obtained upon exposure to bortezomib (Fig S2F), an inducer of ER stress and cell death. In contrast, the proportion of stress-induced cell necrosis was higher in C403A than in CTRL cells (Fig S2F and G). The disorganization in the crista structure is a consequence of OMA1 deficiency (Viana et al, 2021). To define qualitative changes in mitochondrial and crista organization, we performed a transmission electron microscopy (TEM) analysis on in vitro–grown, untreated, or CCCP-stressed cell lines. As shown in Fig 2H, CTRL cell lines illustrated the mitochondrial distribution, electron density, and crista ultrastructure observed under untreated conditions. The number of mitochondria–ER membrane contact sites (MCS) per field of view was quantified (Fig 2I). CTRL cells showed a high proportion of well-structured mitochondria and frequent MCS, up to 8 per field. Upon CCCP treatment, mitochondrial morphology was more heterogeneous and often displayed structural abnormalities in the matrix. In edited cell lines, we observed a 60% reduction in the proportion of MCS and addition of CCCP further exaggerated the accumulation of mitochondrial alterations. We then analyzed the organization of cristae as described in Lam et al (2021). Whereas CCCP treatment reduced the area of cristae in CTRL cells, that of CCCP-treated C403A cells remained significantly larger in agreement with the loss of OMA1-induced mitochondrial fission (Figs 2J and S2H).

### Grafted C403A tumor cells show impaired growth in immunocompetent mice

Metabolic rewiring of mitochondrial activity may have a significant impact on tumor growth (Giessner et al, 2018; Miallot et al, 2021). Because tumor development is often associated with mitochondrial stress (O'Malley et al, 2020), we quantified the growth of CTRL versus C403A cells in nude mice to limit the contribution of adaptive immunity. Both CTRL and C403A grafted clones developed tumors. Although interclonal variability in their growth rate was apparent, it was unrelated to their genotype (Fig 3A). In conclusion, the C403A mutation had no significant impact on their intrinsic growth potential in vivo.

We then quantified OMA-1 expression and proteolytic activity on its targets in tumors grown in nude mice. We performed a Western blot analysis on protein extracts from enriched CD45-negative tumor cells to monitor the expression of OMA1, OPA1, and DELE1 isoforms involved in the handling of mitochondrial and ER stress, respectively (Fessler et al, 2020). As observed with cultured cells, S-OMA1 expression was preponderant in CTRL but almost

undetectable in C403A tumor cells (Fig 3B). Similarly, the 90-kD S-OPA1 isoform predominated over the 120-kD L-OPA1 isoform in CTRL but not C403A tumors (Fig 3B and C). The situation with DELE1 was more heterogeneous between tumors, but overall, the proportion of the S-DELE1 isoform was higher in CTRL versus C403A tumor cells (Fig S3A and B for quantification). To directly test whether C403A OMA1 could contribute to the proteolysis of the 65-kD L-DELE1 into the cytosolic 56-kD S-DELE1 isoform, we exposed cultured cells to oligomycin and monitored the appearance of the S-DELE1 isoform in total cell extracts. As shown in Fig S3C, the L- and S-DELE1 isoforms were detected even in the absence of oligomycin. In C403A cells, the S-DELE1 isoform was predominant in all conditions. This suggested that in mutant tumor cells, C403A OMA1 or other proteases might be able to cleave DELE1 under basal conditions. Altogether, these results demonstrate that the C403A mutation prevents stress-induced OMA1 activation and downstream cleavage of OPA1.

We then tested the growth of various clones grafted in immunocompetent mice. Interestingly, whereas CTRL cells grew exponentially, edited cells failed to expand (Fig 3D). To test the potential of C403A cells to grow in immunocompetent mice, we combined them with CTRL cells and followed the growth of chimeric tumors containing a mix of CTRL and C403A cells. In these experiments, 3 × $10^5$ cells of each type were injected to enhance the detection of small or slowly growing tumors (Fig 3E). Interestingly, the addition of an equivalent number of C403A to CTRL cells slowed down the growth of the latter until day 18. To quantify the relative proportion of CTRL versus C403A cells, we designed a PCR assay able to discriminate their respective contribution to mixed cultures or chimeric tumors (Fig 3F and S3D). On day 12 post-grafting, both CTRL and C403A cells were detected although CTRL cells were more abundant. On day 21, C403A cells were barely detectable in most of the tumors. These results suggest that although C403A cells could grow in immunocompetent mice, their development was rapidly impaired and delayed the growth of CTRL cells, possibly through the release of immunogenic signals.

### OMA1 C403A edition induces a mitochondrial stress associated with the development of anti-tumor immune responses

Immunogenic cues may derive from increased mitochondrial stress within the tumor. We scored mitochondrial polarization and mass, ROS, and mtDNA release in CD45$^-$ cells extracted from CTRL and C403A tumor masses. CD45$^-$ cells from C403A tumors showed significantly reduced mitochondrial polarization and mass (Fig 4A). Whereas total ROS levels were comparable in both types of tumors (Fig 4B), the abundance of cytosolic mtDNA was higher in C403A versus CTRL CD45$^-$ cells (Fig 4C). Dendritic cells but not macrophages can sense tumor mtDNA, a process leading to enhanced

---

and C403 cells after 12-h staurosporine stimulation (n = 2). **(H, I, J)** EM analysis of CTRL and C403A MCA205 cells. Cells were treated for 1 h with CCCP and fixed for TEM acquisition. Scale bars: 200 nm (n = 2). Quantification was performed on 16 independent fields obtained from CTRL and C403A MCA205 cells. **(I)** We scored MCS (panel I) per field in untreated or CCCP-treated conditions. **(J)** Additional analyses were performed to evaluate crista area (panel J). Mann–Whitney test; ****$P$ < 0.0001, ***$P$ < 0.001, and **$P$ < 0.01.

Source data are available for this figure.

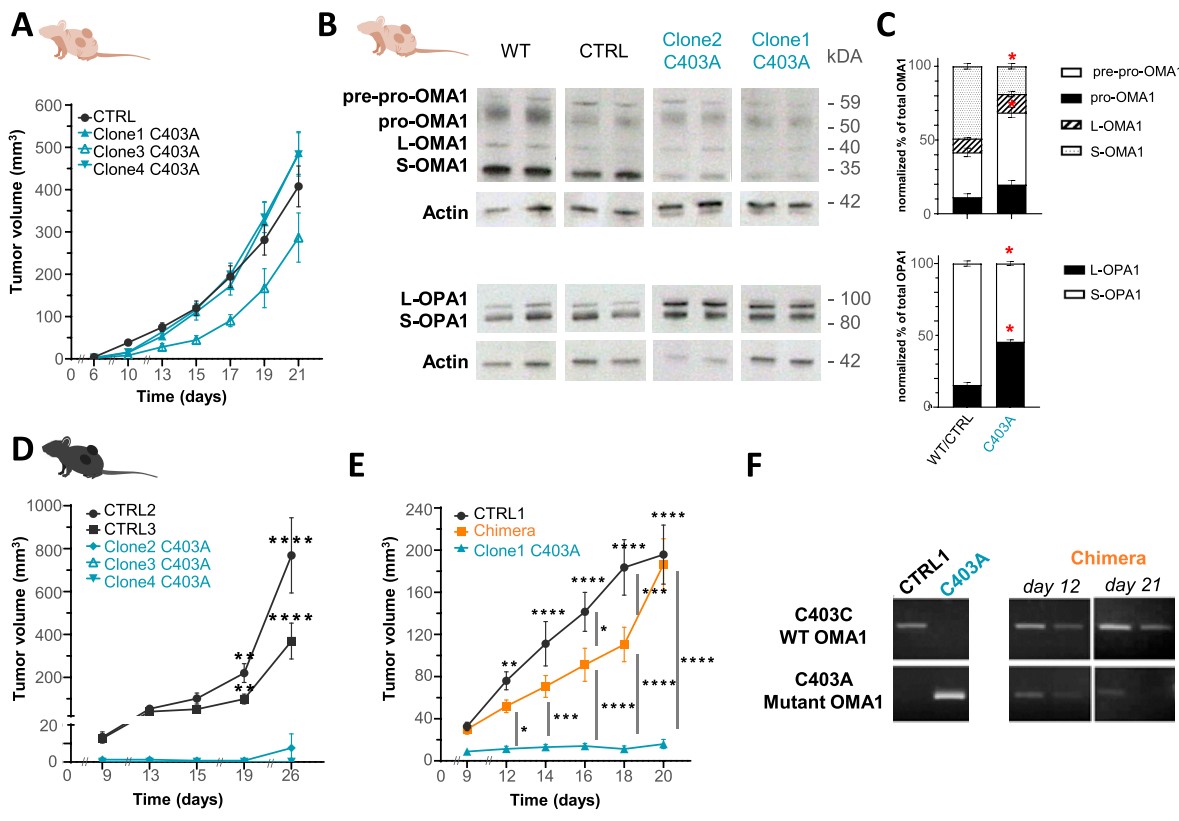

**Figure 3. Growth potential of C403A and CTRL tumor cells in vivo.**
**(A)** Tumor growth in nude mice. $10^5$ WT and C403A MCA205 cells were subcutaneously grafted in the two flanks of mice, and tumor volume was quantified (n = 10). **(B)** Western blot analysis of total protein extracts from CD45-negative cells isolated from WT or C403A tumors at day 21 post-cell engraftment. **(C)** OMA1 and OPA1 expressions were evaluated and quantified in panel (C), as in Fig 2 (n = 4). **(D, E)** Tumor growth in C57BL/6 mice. $3 \times 10^5$ CTRL, $3 \times 10^5$ C403A, or a mix of $3 \times 10^5$ CTRL and $3 \times 10^5$ C403A MCA205 cells were grafted in the two flanks of mice, and tumor volume was quantified (n = 10). Two-way ANOVA with Šídák's multiple comparisons test; ****$P < 0.0001$, ***$P < 0.001$, **$P < 0.01$, and *$P < 0.05$. **(F)** PCR screening to evaluate the proportion of the C403A MCA205 cell in chimera tumors at day 12 and 21 post-cell engraftment (n = 2). Source data are available for this figure.

cross-presentation of tumor antigen (Xu et al, 2017). Furthermore, the XCR1[+] cDC1 cell subset cross-presents antigen and primes anti-tumor CD8[+] cytotoxic lymphocytes (Ferris et al, 2020). To test the contribution of these processes to the control of tumor growth in immunocompetent mice, we grafted CTRL and C403A tumor cells in $Xcr1^{DTA}$ mice that lack cDC1 cells (Wohn et al, 2020). As shown in Fig 4D, CTRL and C403A cells developed tumors at the same rate in these mice; furthermore, the difference in C403A tumor growth between immunocompetent (Fig 3D) and $Xcr1^{DTA}$ mice (Fig 4D) confirmed the major role of DC in sensing tumor-derived stress for the initiation of immune responses.

We then characterized by flow cytometry the immune infiltrate at days 12 (Fig S3E) and 21 (Fig 4E) after grafting. The proportion of CD45[+] cells represented around 40–60% of the tumor mass, but this proportion increased significantly in C403A compared with CTRL tumors. Among CD45[+] cells in non-edited tumors, CD11b[+] myeloid cells represented 40% on day 12 and up to 60% of infiltrating cells on day 21. In C403A tumors, CD11b[−] cells were preponderant and reached 60% of the immune infiltrate. More specifically, CD8[+], CD4[+], and NK1.1[+] cells predominated and were twice more abundant in edited tumors on day 21, whereas the proportion of tumor-associated macrophages was significantly reduced (Fig 4F). We

then tested whether the treatment from days 13 to 20 with an anti-CD8 mAb would rescue the growth of edited tumors. As shown in Fig 4G, the enhancing effect was only transient and most of the edited clones disappeared. This suggested that in addition to intrinsic C403A cell death, CD8[+] T cells and probably other cytotoxic cells contributed to the control of tumor growth.

### High OPA1 expression correlates with poor prognosis in sarcoma subtypes

Results obtained in the mouse model suggest that the loss of mitochondrial adaptation to stress may generate immunogenic signals that enhance anti-tumor immunity. We searched for correlations between OMA1 and OPA1 mRNA expressions and clinicopathological and immune variables in our merged cohort of complex genomic STS. It included 921 clinical samples from non-metastatic and operated primary tumors, comprising 726 samples informative for OMA1 expression and 845 for OPA1. Their characteristics are summarized in Table S1. OMA1 and OPA1 mRNA expressions were heterogeneous across the whole cohort with a range of intensities over 5 and 4 units in the $\log_2$ scale, respectively (Fig 5A), allowing the search for correlations with other variables. No

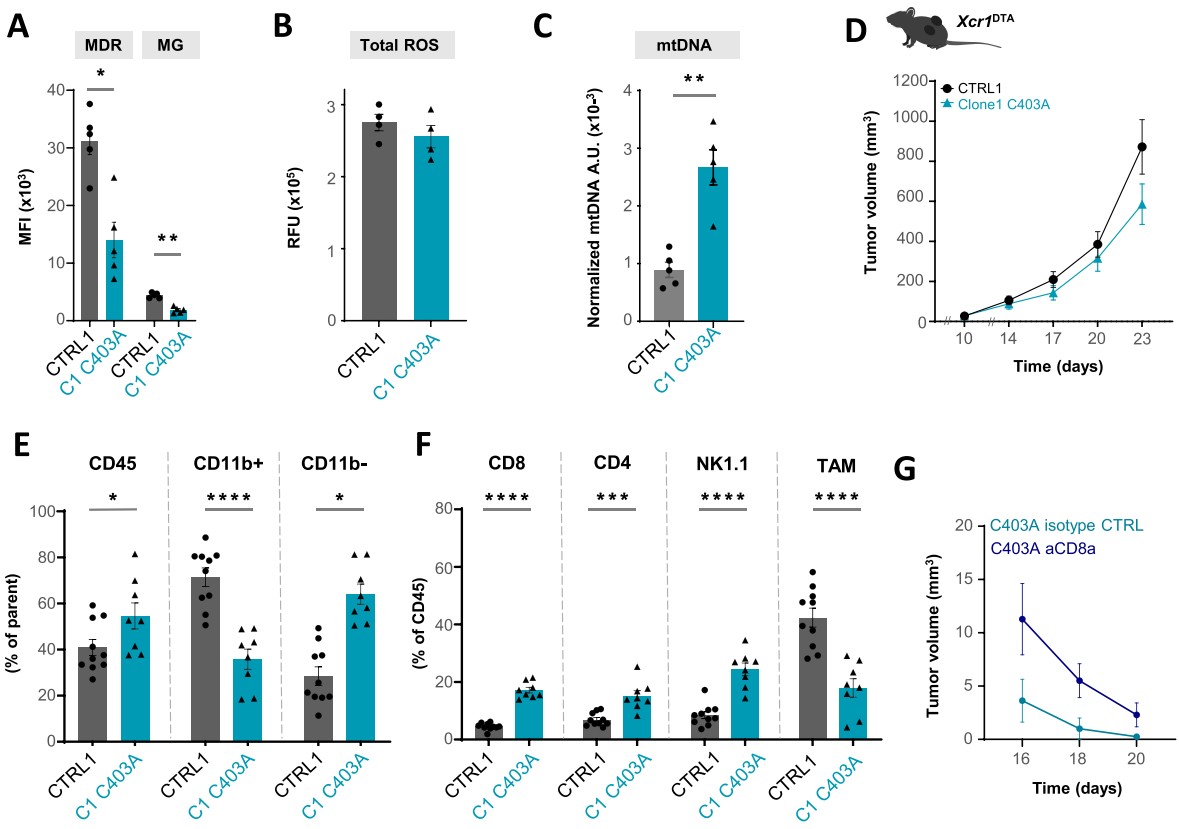

**Figure 4. Evaluation of mitochondrial alterations and immune infiltrate in C403A tumors.**
**(A)** Quantification of mitochondrial depolarization and mass by flow cytometry in CD45-negative cells from WT and C403A MCA205 tumors at day 12. Mann–Whitney test; * $P < 0.05$ and **$P < 0.01$ (n = 4–5). **(B, C)** Relative proportion of total ROS production in CTRL and C403A tumors (n = 4) (C). Relative proportion of cytosolic mtDNA (Nd1) by PCR in CD45-negative cells isolated from CTRL and C403A tumors. Mann–Whitney test; **$P < 0.01$ (n = 5–6). **(D)** Tumor growth in $Xcr1^{DTA}$ mice. $3 \times 10^5$ CTRL or C403A MCA205 cells were subcutaneously grafted in the two flanks of mice, and tumor volume was quantified (n = 8). **(E, F)** Analysis of the immune infiltrate of CTRL or C403A tumors in WT mice at day 21 post-cell engraftment. Mann–Whitney test; **$P < 0.01$ and * $P < 0.05$ (n = 8–10). **(G)** Tumor growth of C403A tumors in CD8 T cell–depleted mice after day 13 (n = 6). Two-way ANOVA with Šidák's multiple comparisons test; *$P < 0.05$.
Source data are available for this figure.

significant correlation existed with patients' age and gender, and pathological grade (Table S1). Correlations were found with the pathological subtype and the tumor site, with more leiomyosarcomas (LMS) and less undifferentiated pleomorphic sarcomas (UPS) among "OMA1-high" tumors than among "OMA1-low" tumors ($P = 6.87 \times 10^{-6}$) and more LMS and less liposarcomas and myxofibrosarcomas among "OPA1-high" tumors than among "OPA1-low" tumors ($P = 9.25 \times 10^{-8}$). OPA1 expression was also associated with the Complexity INdex in SARComa (CINSARC) risk with more "high risk" among "OPA1-high" tumors ($P = 3.53 \times 10^{-3}$). Because OMA1 regulation of activity depends on post-translational modifications, variations in OMA1 transcripts might have a modest impact on prognosis as confirmed by the analysis of metastasis-free survival (MFS) in OMA1-high or OMA1-low patients (Fig 5B). The 5-yr MFS was 48% (95% CI 40–57) in the "OMA1-high" class versus 59% (95% CI 53–67) in the "OMA1-low" class ($P = 0,062$, log-rank test; Fig 5B). In contrast, variations in OPA1 levels might ultimately tune mitochondrial fusion potential. Accordingly, the prognostic analyses (Fig 5C) showed that an "OPA1-high" status was associated with reduced MFS in univariate ($P = 1.11 \times 10^{-2}$, Wald's test) and multivariate (hazard ratio (HR) = 1.35, 95% CI 0.96–1.91, $P = 0.089$)

analyses (Table S2), and 5-yr MFS was 48% (95% CI 40–57) in the "OPA1-high" class versus 60% (95% CI 52–69) in the "OPA1-low" class ($P = 5.48 \times 10^{-3}$, log-rank test; Fig 5C).

Then, we investigated whether OPA1 expression was associated with immune variables in our clinical samples (Fig 5D). First, we compared the composition and functional orientation of tumor-infiltrated immune cells using the 24 immune cell types defined as the immunome. Significant differences existed between "OPA1-high" and "OPA1-low" classes; "OPA1-low" tumors displayed a higher infiltrate than "OPA1-high" tumors in 12 immune cell types ($P < 0.05$) including T cells, Tem cells, Th1 cells, Th17 cells, cytotoxic cells, CD56dim NK cells, dendritic cells (DC, iDC, aDC, and pDC), macrophages, and neutrophils. Second, "OPA1-low" tumors displayed higher expression of many signatures related to antigen presentation (Table S3). Finally, "OPA1-low" samples displayed a higher immune cytolytic activity score ($P = 4.39 \times 10^{-6}$) and ICR score ($P = 1.60 \times 10^{-5}$), which reflect an anti-tumor cytotoxic immune response, than "OPA1-high" samples, and higher scores for signatures associated with response to immune checkpoint inhibitors (ICI): T cell–inflamed signature (TIS) ($P = 2.86 \times 10^{-6}$) and tertiary lymphoid structure score ($P = 5.66 \times 10^{-7}$). Altogether, these results

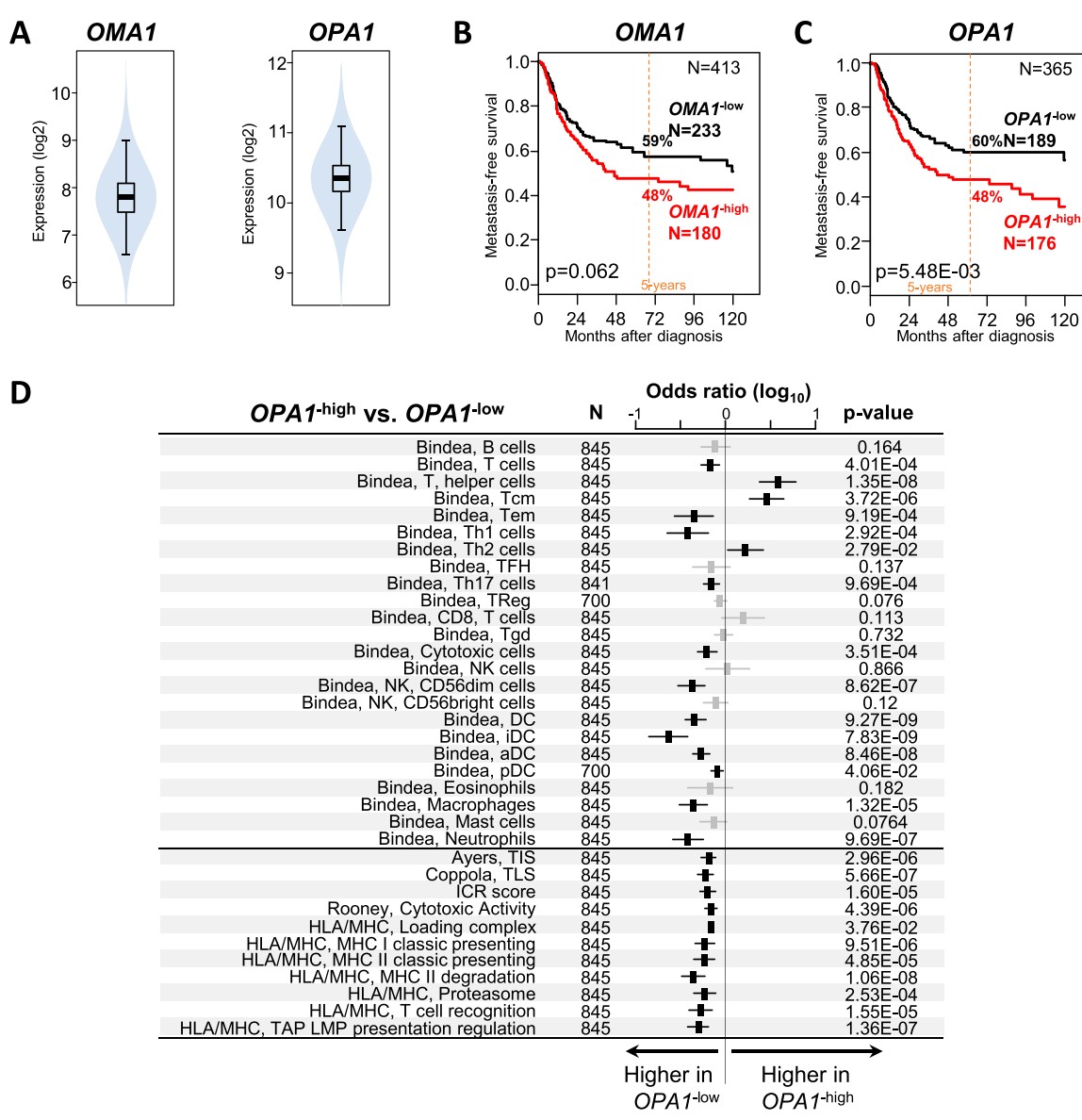

**Figure 5. OMA1 and OPA1 expressions in clinical samples of STS with complex genomics.**
**(A)** Violin plots showing the distribution of mRNA expression levels of OMA1 and OPA1 in 921 tumor samples. **(B, C)** Kaplan–Meier MFS curves according to OMA1 and OPA1 expressions. The *P*-values are for the log-rank test. **(D)** Correlations between OPA1 expression-based classification and immune variables. Forest plots of correlations between OPA1-high and OPA1-low expressions and immune features including the 24 Bindea's innate and adaptive immune cell subpopulations, the TIS signature and the TLS signature associated with response to immune checkpoint inhibitors, the ICR score and the cytolytic activity score associated with anti-tumor cytotoxic immune response, and several antigen-processing signatures. The *P*-values are for the logit link test.

obtained in complex genomic STS reinforce the notion that variations in OMA1/OPA1 expression might condition the development of immune responses.

# Discussion

Tumor development is progressively associated with mitochondrial stress that triggers several compensatory mechanisms (O'Malley et al, 2020). Preservation of mitochondrial fitness depends on the induction of cytoprotective mitochondrial proteins via the retrograde

mitochondria–nuclear signaling and on the tight balance between fusion and fission (da Cunha et al, 2015). Ultimately, damaged mitochondria are scavenged by mitophagy. Through the regulation of its proteolytic activity, the OMA1 metalloprotease is a major sensor of mitochondrial stress (Ehses et al, 2009; Baker et al, 2014). A redox molecular switch involving a disulfide bridge between two cysteines has been identified in yeast OMA1 (Bohovych et al, 2019). We confirmed in murine sarcoma cells that the mutation of cysteine 403 abrogated OMA1 proteolytic activity toward the OPA1 protein and also possibly its autocatalytic activity. Indeed, the production of the long and short OMA1 isoforms is the result of a complex interplay between

the YMEL1 and OMA proteases (Wang, MCB 21). In contrast, we could not obtain conclusive results concerning DELE1 processing by C403A OMA1. Although the role of this disulfide bridge is not fully understood, results in yeast suggest that it may participate in OMA1 stability and/or interaction with other proteins (Bohovych et al, 2019). Indeed, OMA1 participates in the stability (Viana et al, 2021) and function of the MICOS complex in the control of cell bioenergetics and inter-membrane contacts (Bohovych et al, 2015; Sakowska et al, 2015; Burke, 2017; Wollweber et al, 2017; Viana et al, 2021). Furthermore, the MICOS complex is sensitive to variations in the oxidation status of several proteins including OMA1 (Bohovych et al, 2019) and MIC19 (Sakowska et al, 2015). Although we could not detect the presence of OMA1 in the complexes because of the lack of appropriate antibodies, the absence of cysteine 403 did not impair complex formation but prevented the proteolysis of OPA1 within the MICOS complex. OPA1 is positioned upstream of MICOS and regulates crista junction width (Glytsou et al, 2016) and cell death (Burke, 2017). Interestingly, the C403A mutation had a functional impact on mitochondrial organization, leading to the accumulation of abnormal cristae, depolarized mitochondria, reduced MCS, and mitochondrial ATP production.

Our results show that the C403A mutation does not impair tumor cell growth in immunodeficient mice. Growth depends on the metabolic rewiring occurring in the competitive tumor environment. C403A cells showed a preferential use of glycolysis over respiration for ATP production. In vivo, this phenotypic bias should favor the development of a Warburg phenotype that enhances tumor aggressiveness, particularly in the absence of a potent anti-tumor immunity (Giessner et al, 2018). The modalities of cell death are known to influence tumor immunogenicity (Kroemer et al, 2013). Interestingly, C403A cells, while more resistant to apoptosis induction, showed increased stress-induced cell death in vitro associated with mtDNA release in the cytosol. Because S-OPA1 contributes to the maintenance of mtDNA and cristae (Del Dotto et al, 2017), the inability to maintain a correct proportion between the L-OPA1 and S-OPA1 isoforms in C403A cells may explain this result. mtDNA is a major immunostimulating danger-associated molecular pattern, which participates in endocellular inflammasome-mediated (Nakahira et al, 2011; Tschopp, 2011; Galluzzi et al, 2012; Zhong et al, 2018) or extracellular neutrophil-mediated (Zhang et al, 2010; Oka et al, 2012) triggering of inflammation. The consequences of mtDNA release in tumor microenvironment vary depending on the nature of infiltrating immunocytes, associated with the neutrophil-mediated worsening of tumor progression (Singel et al, 2019) or, reciprocally, with the stimulation of the cross-priming potential of cDC1 cells for the development of anti-tumor cytotoxic CD8 cells (Xu et al, 2017). Our results support this latter hypothesis because over time, edited tumors were enriched in NK1.1$^+$ and CD8$^+$ lymphocytes, and furthermore, in the absence of cDC1 cells, C403A clones grew in immunocompetent mice. In addition, the presence of mutant tumor cells in a chimeric tumor slowed down the growth of the non-edited clones and was associated with the accumulation of an immune infiltrate.

The relevance of our findings in a mouse sarcoma model was questioned by exploring the levels of expression of the transcripts coding for OMA1 and OPA1 in a large series of STS with complex genomics, documented in various databases. Whereas the level of expression of these proteins is relatively independent from their activation status, few contradictory studies reported an association of OMA1 levels with susceptibility to cancer (Alavi, 2019). The OPA1 protein regulates tumor growth through the modulation of angiogenesis and apoptosis, but its contribution to tumor cells themselves has not been documented (Herkenne & Scorrano, 2020). Our results in sarcoma suggest that the expression level of OMA1 and OPA1 varies significantly among sarcoma subtypes with complex genomics. This may reflect an adaptation to mitochondrial stress in tumor clones allowing the selection of the fittest variants. Furthermore, the complex functions of various OPA1 isoforms on the control of crista structure and fission would justify to complete our study with a biochemical analysis of OPA1 protein on tumor samples. The most significant observation was to show the good prognosis value of low OPA1 expression for MFS in STS and the associated presence of immune signatures linked to IFN-γ signaling, MHC expression, and infiltration by immune cells with potential anti-tumor functions. This type of signature evokes that found in the OMA1 mutant model that was developed in which stress-induced OPA1 cleavage is prevented, limiting adaptation to mitochondrial stress of tumor cells and exposing to increased immunogenic cell death. One might hypothesize that a reduced level of OPA1 could lead to a similar result under stress conditions. One report showed that the use of an OPA1 inhibitor (MYLS22) could limit tumor growth (Herkenne et al, 2020). However, in this report, the authors concluded in favor of angiogenic alterations. Because the effect of this inhibitor in vivo is not clarified, the interpretation of these results must be cautious. Based on our results, one could propose to design modulators of OMA1 or OPA1 activation targeting the redox switch or the fusiogenic function of OPA1 to enhance tumor cell fragility and immunostimulation.

# Materials and Methods

### Animals

8–10-wk-old female C57BL/6 and NMRI-nu mice were purchased from Janvier Laboratories. Few experiments were performed on male mice. Xcr1$^{Cre-mTFP1}$ and Rosa$26^{LSL-DTA}$ (B6.129P2-Gt(ROSA)26Sor$^{tm1(DTA)Lky/J}$) were previously described (Voehringer et al, 2008; Wohn et al, 2020). Xcr1$^{Cre-mTFP1}$ mice were crossed to Rosa$26^{lsl-DTA}$ mice in which Cre-mediated excision of a loxP-flanked transcriptional STOP element triggers the expression of diphtheria toxin fragment A (DTA), and results in the constitutive ablation of cDC1 in Xcr1$^{DTA}$. Mice were housed under a standard 12-h:12-h light–dark cycle with ad libitum access to food and non-acid water, 22°C ± 1°C, and 45–60% humidity, and were maintained under specific pathogen-free conditions at the animal facility of the Centre d'Immunologie de Marseille-Luminy (F1305510). Experimentations were authorized by the Ethical Committee for Animal Experimentation (#30566–2021032215496999 v2; APAFIS). Collaborative experiments using Xcr1$^{DTA}$ mice (Wohn et al, 2020) were performed in B. Malissen's laboratory (#26488–2020070612584424 v2; APAFIS).

### Cell lines

The MCA205 cell line was kindly provided by E Vivier at Centre d'Immunologie de Marseille-Luminy and cultured in DMEM/F-12

(Gibco) supplemented with 10% FBS (Gibco), 100 μg/ml penicillin–streptomycin (Gibco), 2 mM of L-glutamine, and 1 mM of sodium pyruvate at 37°C with 10% $CO_2$. Mitochondrial stress was triggered by incubating cells in 20 μM CCCP for 1 h at 37°C. Mycoplasma status was checked using MycoAlert Lonza Detection Kit, and cells were used at low passage. For cell number quantification, $10^5$ cells were seeded at low density in a 12-well plate, and cells were harvested every 2 d with trypsin and counted with a cell counter (CASYton).

### Edition of the *Oma1* gene

Mutated C403A clones were obtained using the prime editing (PE) technique as described (Anzalone et al, 2019). CTRL OMA-1 corresponds to cells that were subjected to the editing process but did not internalize the mutation. PE (#132775; Addgene) produces template-directed local sequence changes in the genome without the requirement for DSBs or exogenous donor DNA templates. We designed the pegRNA sequences using the pegFinder online tool (Chow et al, 2021) to target the desired genomic sequence. Cells were first edited using the PE2 system that edits only one DNA strand and is expected to have a maximum editing efficiency of 50%. To obtain homozygous mutants, heterozygote clones were submitted to the PE3 editing system that uses an additional sgRNA to direct SpCas9H840A to nick the non-edited DNA strand and encourages the edited strand to be used as a repair template by DNA repair factors, leading to a further increase in editing efficiency.

Plasmids expressing pegRNA were constructed by Golden Gate assembly. Sequences of sgRNA and pegRNA are listed in the Supplemental Data 1. The oligonucleotides corresponding to the pegRNA spacer, pegRNA 3′ extension, and pegRNA scaffold were annealed and assembled into the BsaI-digested pU6-pegRNA-GG acceptor vector (#132777; Addgene). For the PE3 editing system, the sgRNA was cloned in the pLKO.1-puro-GFP vector (Phelan et al, 2018). All vectors for mammalian cell experiments were purified using EndoFree Plasmid Maxi Kit (QIAGEN). The pCMV-PE2 and the pU6-pegRNA-GG-acceptor were a gift from David Liu (Addgene plasmid #132775; http://n2t.net/addgene:132775; RRID:Addgene_132775; Addgene plasmid #132777; http://n2t.net/addgene:132777; RRID: Addgene_132777).

$8 × 10^5$ MCA205 cells were seeded in six-well plates, and transfections were conducted when cells reached ~70% confluency after 16–20 h. Cells were transfected with jetOPTIMUS reagent (Polyplus) following the manufacturer's instructions. For PE2 experiments, cells were transfected with 2 μl of jetOPTIMUS, 1,5 μg of PE2 plasmid (#132775; Addgene), 500 ng of pegRNA plasmid (#132777; Addgene), and 200 ng of peGFP-C1 vector plasmid (Clontech). The transfected cells were collected after 72 h of culture and GFP-positive cells sorted in single clones in 96-well plates. After genomic DNA extraction (QIAGEN), clones were screened by PCR (Supplemental Data 2) for the presence of mutation. 12% of the clones were positive for the mono-allelic mutation and confirmed by sequencing (Fig S1E). For PE3, cells were transfected with 2 μl of jetOPTIMUS, 1,5 μg of PE2 plasmid, 500 ng of pegRNA plasmid, and 200 ng of pLKO-GFP-sgRNA plasmid. The transfected cells were collected after 72 h of culture and GFP-positive cells sorted in single

clones in 96-well plates. Clones were screened by PCR for the presence of mutation and the absence of WT sequence. Four of 40 clones presented the bi-allelic mutation, confirmed by sequencing.

### Tumor experiments

MCA205 cells ($10^5$ or $3 × 10^5$ depending on experiments) were subcutaneously grafted in the flanks of C57BL/6, $Xcr1^{DTA}$, or nude mice. For tumor growth monitoring, mice were anesthetized with 2.5% isoflurane every 2 d. Tumor size was assessed with a caliper by measuring the length (L) and width (W) of the tumor. Tumor volumes were calculated using the following formula: $(L × W)^2/2$. A limit point was settled when tumor volume was above 1,000 $mm^3$. Tumors were harvested between D10 and D26 post-implantation, each tumor being considered as an experimental unit referred to as n. Animals were euthanized when severe bleeding or scars were detected on the tumor implantation site or when they presented symptoms of poor health (weight loss, prostration). In vivo $CD8^+$ cell depletion was achieved by injecting the purified anti-CD8 mAb (clone 53–5.8) intraperitoneally from day 13 and every 3 d at 200 μg per mouse, respectively. Rat-IgG1 anti-horseradish peroxidase (clone HRPN) was used as an isotype CTRL after the same dosage.

Tumors were mechanically and enzymatically digested using the Miltenyi Biotec gentleMACS Octo Dissociator technology. Samples were filtered through a 70-μm Cell Strainer (Becton Dickinson) to remove cell clumps and submitted to red blood cell lysis (eBioscience buffer). Cell suspensions were analyzed by flow cytometry. Alternatively, CD45-negative cells were isolated using CD45 microbeads from Miltenyi Biotec and the MultiMACS Cell24 separation on LS columns according to the manufacturer's protocols.

### Analysis of OMA1 and OPA1 expressions in soft tissue sarcoma clinical samples

We analyzed our database (Bertucci et al, 2022) including clinico-pathological and normalized gene expression data of clinical STS samples gathered from 16 public data sets (Table S4). All samples were from an operative specimen of previously untreated primary tumors. The gene expression profiles had been generated using DNA microarrays or RNA sequencing. Because our mouse model represented a sarcoma with complex genomics, our analysis was limited to the 921 cases of STS with complex genomics. The most frequent pathological types were LMS and UPS, and 56% of cases were high-risk CINSARC. *OMA1* and *OPA1* mRNA expressions were analyzed as discrete variables (high versus. low) using their mean expression level of the whole series as cutoff. Based on the link we observed in our mouse model between OMA1/OPA1 and immunity, we searched for correlations between OPA1 tumor expression and immunity-related variables. These latter were represented by the following multigene classifiers/scores: the 24 Bindea's innate and adaptive immune cell subpopulations (Bindea et al, 2013), several antigen-processing machinery signatures (Tables S1, S2, and S4), two signatures associated with anti-tumor cytotoxic immune response (the Immunologic Constant of Rejection classifier (Bertucci et al, 2018) and the cytolytic activity score (Rooney et al, 2015)), and two metagenes associated with response to immune checkpoint inhibitors (the T cell–inflamed signature (Ayers et al, 2017) and the

tertiary lymphoid structure signature (Coppola et al, 2011)). We also applied the CINSARC signature, now recognized as the most relevant prognostic signature in STS (Chibon et al, 2010) that identifies patients as either high risk or low risk of relapse. The correlations between OMA1 or OPA1 expression-based classes and clinico-pathological variables and molecular signatures were measured using Fisher's exact test or a *t* test when appropriate. The endpoint of prognostic analysis was the MFS, calculated from the date of diagnosis until the date of metastatic relapse or death from any cause, whichever occurred first. The follow-up was measured from the date of diagnosis to the date of last news for event-free patients. Survival rates were estimated using the Kaplan–Meier method and curves compared with the log-rank test. Uni- and multivariate prognostic analyses were done using the Cox regression analysis (Wald's test). The variables tested in univariate analysis were the patients' age and gender, pathological tumor type (UPS, LMS, pleomorphic liposarcomas, myxofibrosarcomas, others), pathological grade (2–3, 1), tumor site (extremities, head and neck, internal trunk, superficial trunk), CINSARC-based risk (high, low), and the OMA1- or OPA1-based classification (high, low). Multivariate analysis incorporated all variables with a *P*-value inferior to 5% in univariate analysis. The correlations between molecular immune variables and OPA1-based classification were assessed by logistic regression analysis with the glm function (R Statistical Package; significance estimated by specifying a binomial family for models with a logit link). All statistical tests were two-sided, and the significance threshold was 5%. Analyses were done with the survival package (version 2.43) from R software (version 3.5.2).

## Flow cytometry

Cell death was evaluated using the LIVE/DEAD Fixable Blue flow cytometry, and unspecific labeling was prevented by incubation with anti-CD16/CD32 mAb in PBS/2 mM EDTA for 30 min. Cell surface antibody labeling was performed in FACS buffer for 1 h at 4°C. For intracellular staining, the Foxp3/Transcription Factor Staining Buffer Set (eBioscience) was used. Cells were analyzed on BD LSR Symphony or Fortessa (Becton Dickinson). Data analysis was done using FlowJo 10.8 software. To evaluate mitochondrial depolarization and mitochondrial mass, cells were stained with MDR (10 nM) and MitoTracker Green (100 nM) for 20 min at 37°C (Thermo Fisher Scientific) in serum-free RPMI. For staining of cultured cells, we used 100 nM MDR. CD8 and CD4 T cells were quantified within CD45[+] CD11b[−] NK1.1[−] cells, and tumor-associated macrophages, within CD11b[+] Ly6G[−] NK1.1[−] Ly6C[low] MHCII[+] CD64[+] F4/80[+] cells.

## Annexin V/SYTOX Blue assay

$3 \times 10^5$ CTRL and C403A cells were seeded in a six-well plate. After 24 h of 1 µM staurosporine (used as a positive control) or 1 µM bortezomib stimulation, cells were harvested and stained for 20 min with 1 µM SYTOX Blue in PBS. Apoptotic cells were stained with Annexin PE for 20 min and immediately analyzed on FACSCalibur/Canto II. Data analysis was done using FlowJo 10.8 software.

## Recombinant protein production

We produced a truncated version of OMA1 protein starting on amino acid 213 (SPVTGR …) corresponding to the beginning of the outer membrane region. The corresponding sequence carrying or not the C403A mutation in a synthetic gene (codon optimized for *E. coli* expression) was cloned into a prokaryotic expression vector. The OMA1 fusion proteins produced in *E. coli* contain a signal sequence to export the protein to the *E. coli* periplasmic space, followed by a hexahistidine tag for purification, a protein disulfide-isomerase (DsbC) fusion to help disulfide bridge formation, and a TEV recognition sequence to allow cleavage and isolation of the OMA1 alone if necessary. The gene synthesis and cloning were outsourced (Twist Bioscience). For the production phase, the proteins were transformed in T7 express strains (NEB) and grown in 400 ml TB media for 24 h at 25°C (induction with 1 mM when O.D was at 0.8). Cells were centrifuged and frozen in purification buffer A (Tris 50 mM, NaCl 300 mM, and imidazole 10 mM, pH 8) with the addition of 0.5 mg/ml of lysozyme. After thawing, the lysed bacterial pellets were treated with DNase, sonicated (6*30 s), and centrifuged for 40 min at 20,000*g*. Supernatants were run on AKTA xPress on a 5-ml Nickel HisTrap FF crude column (Cytiva), washed in buffer A with 50 mM imidazole, and eluted in a buffer containing 250 mM imidazole. The eluted material was dialyzed in PBS overnight and purity-confirmed by SDS–PAGE analysis. A protein with the expected molecular weight was obtained only for the WT sequence and not the mutated version that could not be purified. The purified OMA1 protein (His-DsbC) was used for enzymatic characterization.

## Structure prediction

The mouse OMA-1 amino acid sequences were obtained from ENSEMBL (ENSMUSG00000035069). With the advent of powerful AI implementations, the AlphaFold Protein Structure Database (https://alphafold.ebi.ac.uk/) and DeepMind's Colab notebook (Jumper et al, 2021) have been used for structure prediction. Figures representing structural renderings were generated with the PyMOL Molecular Graphics System (version 2; Schrödinger, LLC).

## Electron microscopy

The cells were fixed with 2.5% glutaraldehyde in 0.1 M cacodylate buffer for 30 min. After three washes over 15 min in the same buffer, the cells were post-fixed with 1% OsO4 in the same buffer. After three washes over 15 min in water, the cells were dehydrated in 50% ethanol for 10 min and 70% ethanol for 10 min and incubated in uranyl acetate 2% in 70% ethanol for 30 min. Dehydration was then pursued with a single bath of 95% ethanol, three baths of pure ethanol, and three baths of acetone (10 min each). The cells were then infiltrated with Epon resin in acetone (1:2, 2:2, 2:1, and pure resin, 1 h each) and pure resin overnight. The next day, the pellets were embedded in fresh pure Epon resin and cured for 48 h at 60°C. 70-nm ultrathin sections were performed on a Leica UCT Ultramicrotome (Leica) and deposited on formvar-coated slot grids. The grids were contrasted using lead citrate and observed in a FEI Tecnai G2 at 200 KeV. Acquisition was performed on a Veleta camera (Olympus).

## BN-PAGE

CTRL and C403A OMA1 mutant cells were stimulated with 20 μM CCCP or DMSO control for 1 h. Mitochondria were purified using the mitochondrial isolation kit for cultured cells (89874; Thermo Fisher Scientific) according to the manufacturer's instructions. Samples were lysed in NativePAGE sample buffer with 2% digitonin. 75 μg or 20 μg of non-denatured mitochondrial proteins was prepared with NativePAGE 5% G-250 Sample Additive, separated on precast NuPAGE 3–12% Bis–Tris Mini Protein Gel, and transferred to PVDF membrane. Protein ladder was revealed with Imperial Blue staining, and membranes were immunoblotted with the indicated antibodies.

## Immunoblotting

Cell lysates were prepared in ice-cold RIPA buffer, supplemented with a protease inhibitor cocktail. Samples were centrifuged at 10,000$g$ for 5 min at 4°C and supernatants collected. Protein concentrations were determined using the Pierce BCA assay. Laemmli buffer was supplemented with $\beta$-mercaptoethanol and incubated for 5 min at 95°C. 5 μg of total proteins was loaded on a 4–12% SDS–PAGE run at 150 V and then transferred to PVDF membranes during 1 h at 100 V. Membranes were saturated in PBS/Tween/5% BSA overnight at 4°C, followed by incubation with either mouse anti-OMA-1, anti-OPA1, anti-DELE1, anti-TOM20, and anti-actin antibodies at 4°C (see Supplemental Data 1). Membranes were washed and incubated with HRP-linked goat anti-rabbit IgG or HRP-linked goat anti-mouse IgG for 1 h at RT. The antigen–antibody complex was detected using ECL, according to the manufacturer's instructions. Images were captured on autoradiography films and scanned using Samsung Digital Presenter with a 720P HD document camera with a 14× optical zoom and 3× digital zoom. Signals were quantified with ImageJ software and normalized using actin signal intensity as a reference.

## Enzymatic activity

The OMA1 activity assay relies on a fluorogenic 8-mer peptide derived from the OPA1 sequence containing the OMA1 cleavage site (Ishihara et al, 2006). Using a final 200 μl reaction volume, the reagents were quickly added in the following order in a black 96-well plate: (1) OMA1 activity assay buffer (50 nM of Tris–HCl, pH 7.5, and 40 mM of KCl); (2) 5 μg of protein sample with or without 200 μM zinc chelator N,N,N′,N′-tetrakis(2-pyridylmethyl)ethylenediamine (TPEN); and (3) the OPA1 fluorogenic reporter substrate (5 μM). Relative fluorescence was recorded (excitation and emission, respectively, at 320 and 405 nm) every 5 min for 30 min at 37°C using a fluorescent plate reader (TECAN). For statistical analysis, the average fluorescence of the OPA1 fluorogenic reporter substrate alone was measured (<200 relative fluorescence units).

## Immunofluorescence analysis

C403A and CTRL MCA205 cells were seeded in eight-well Nunc Lab-Tek plates (Thermo Fisher Scientific) and labeled with MDR at 100 nM at 37°C in prewarmed DMEM/F-12 without phenol red for 30 min.

Airyscan imaging was performed using a commercial Zeiss confocal microscope LSM 880 equipped with an Airyscan module (Carl Zeiss AG), and images were taken with a 63x/1.40 NA M27 Plan-Apochromat oil objective. In this mode, the emitted light was projected onto an array of 32 sensitive GaAsP detectors, arranged in a compound eye fashion. MDR-labeled samples were excited with a 633-nm beam, and emission was recorded using BP 570–620 + LP 645 filters. Images were processed using Zen Black 2.3 software. For TOMM20 staining, cells were seeded on glass coverslips, treated with 20 μM CCCP for 45 min, then fixed in PFA 4% for 10 min, and permeabilized in PBS/3% Triton/10% DKS (donkey serum) for 30 min. Mitochondria were stained overnight using a rabbit anti-TOMM20 antibody (186734-1/500; Abcam) and revealed with Alexa Fluor 594–AffiniPure Donkey Anti-Rabbit IgG. Coverslips were prepared using DAPI-containing Fluoromount. The Airyscan processing performs filtering, deconvolution, and pixel reassignment to improve SNR. The filtering (the Wiener filter associated with deconvolution) was set to the default filter setting of 6.1 in 2D. Microscopy images were analyzed using Imaris *5.104* software. To model MDR-labeled mitochondria in 3D, we used the plugin "structure." The threshold was positioned at the curve inflexion point. The mitochondrial volume of fragmented mitochondria varies between 0.08 and 0.53 $\mu m^3$, and that of fused mitochondria is set over 0.53 $\mu m^3$. To quantify fission potential, we calculated the mitochondrial fission index as $\log_{10} (\sum 0.08 \, \mu m3 < Vmit < 0.53 \, \mu m3 / \sum Vmit > 0.08 \, \mu m3)$.

## Holotomographic microscopy

The holotomographic microscopy—Nanolive CXA—allows to capture the real, kinetic response of cells without imaging-induced artifacts. CTRL and C403A cells were recorded during 16 h with an interval of 25 min in the presence or in the absence of staurosporine 1 $\mu M$ diluted in DMSO (final concentration 0.1% DMSO in regular medium condition). The control condition was performed in the presence of DMSO 0.1% diluted in regular medium condition. All the captures were analyzed with the LIVE Cell Death Assay module. The results are expressed by the percentage of apoptosis over time. Experiments were performed in duplicate.

## Seahorse

Agilent Seahorse XFp Real-Time ATP Rate Assay Kit was used to measure OCR and ECAR using a Seahorse XFp Extracellular Flux Analyzer. Cells were seeded at 5 × $10^4$ cells/well for 16 h at 37°C in 10% CO2. One hour before measurement, the cell culture medium was replaced with Seahorse DMEM, pH 7.4, supplemented with 10 mM of glucose, 1 mM of pyruvate, and 2 mM of glutamine, and the miniplate was incubated for 1 h at 37°C in a non-CO2 incubator. Cells were stimulated with 1.5 $\mu M$ of oligomycin and 0.5 $\mu M$ of rotenone/antimycin A. The ATP rate index corresponds to the mitoATP production rate divided by glycoATP production rate at a given time point.

## qRT–PCR analysis

Total mRNA from cells was purified using RNeasy Mini Kit (QIAGEN). For qRT–PCR analysis, 0.5 μg RNA was reverse-transcribed with the

SuperScript II RT kit (Life Technologies). Amplification was performed on a 7,500 Fast Real-Time PCR system (Applied Biosystems) using SYBR Green Master Mix (Takara) and specific primer pairs. Expression levels were normalized to the control gene actin.

### Quantification of cytosolic mitochondrial DNA

To evaluate the content of cytosolic mitochondrial DNA, WT or C403A OMA1 tumors were harvested at day 12 post-cell engraftment and processed as described. Two tumors from the same mice were pooled, and CD45-negative fractions were split into two equivalent fractions and processed for total or cytosolic DNA extraction. For total DNA extraction, 50 mM NaOH was added and cells were incubated at 95°C for 15 min. The reaction was stopped by adding 1 M of Tris–HCl, pH 8. For cytosolic DNA extraction, cells were incubated with cytosolic extract buffer (150 mM NaCl, 50 mM Hepes, and 25 mg/ml digitonin) for 10 min on ice. Cells were centrifuged at 410$g$ for 5 min. The supernatant was centrifuged at 16,000$g$ for 10 min. The total and cytosolic extracts were then used for DNA extraction using DNA Blood & Tissue Kit. Total DNA abundance was evaluated using the Qubit assay. Nd1 (mitochondrial DNA) and POLG1 (nuclear DNA) expression was quantified using ONEGreen Fast qPCR Premix on a 7,500 Fast Real-Time PCR system (Applied Biosystems). For each gene, cycle threshold (Ct) values were obtained on cytosolic and total DNA fractions and their ratio normalized using the amount of total DNA as shown in the formula: $\frac{\frac{Ct\,Nd1\,cytosol}{Ct\,Nd1\,total}}{\frac{Ct\,Polg1\,cytosol}{Ct\,Polg1\,total}} \times \frac{1}{DNA\,abundance}$.

### ROS assay

Total CTRL and C403A tumors were snap-frozen, and total ROS production was evaluated using the OxiSelect In Vitro ROS Assay kit according to the manufacturer's instructions. Briefly, frozen tumors were homogenized on ice. Catalyst was added to cell lysates or hydrogen peroxide standards, mixed, and incubated for 5 min. DCFH (2′,7′-dichlorodihydrofluorescein) solution was added to each well, followed by a 45-min incubation at room temperature. Accumulation of ROS in tumors was calculated by monitoring the fluorescence intensity at excitation and emission wavelengths of 480 and 530 nm.

### Statistics analysis

Sample size was designed to minimize the number of individual experimental units (mice or samples), and obtain informative results and appropriate material for downstream analysis. This represents five mice per group, and experiments were typically performed 2–3 times as stated in figure legends. Error bars represent the standard error of the mean. GraphPad Prism 9 software was used for statistical significance assessment. The Gaussian distribution was tested using the D'Agostino–Pearson omnibus normality test. When passing the normality test, a $t$ test was used. Otherwise, a Mann–Whitney $U$ test was used. Differences were considered to be statistically significant when ****$P$ < 0.0001, ***$P$ < 0.001, **$P$ < 0.01, and * $P$ < 0.05.

## Supplementary Information

## Acknowledgements

The INCA PLBIO19-015 grant supported R Miallot funding and the work realized in the teams of P Naquet and J-Y Blay. We thank the Canceropôle Provence-Alpes-Côte d'Azur, the National Cancer Institute, and the CRISPR Screen Action for the support brought to the realization of this research project. We thank the core facilities involved in this project: CIML flow cytometry, CIML animal facilities, ImagImm (Roxane Fabre), the French Infrastructure for Integrated Structural Biology (FRISBI) ANR-10-INSB-05-01, and the Plateforme d'Imagerie Vasculaire et de Microscopie Intravitale at the C2VN. The electron microscopy experiments were performed on the PiCSL-FBI core facility (IBDM, AMU-Marseille), a member of the France-BioImaging National Research Infrastructure (ANR-10-INBS-0004). BM and SH were supported by DCBIOL LabEx (grants ANR-11-LABEX-0043 and ANR-10-IDEX-0001-02 PSL).

### Author Contributions

R Miallot: conceptualization, formal analysis, supervision, funding acquisition, investigation, methodology, project administration, and writing—original draft, review, and editing.

V Millet: resources, formal analysis, investigation, methodology, writing—original draft, and project administration.

Y Groult: resources, methodology, and project administration.

A Modelska: methodology.

L Crescence: methodology.

S Roulland: methodology.

S Henri: resources.

B Malissen: resources.

N Brouilly: methodology.

L Panicot-Dubois: methodology.

R Vincentelli: resources, funding acquisition, and methodology.

G Sulzenbacher: resources, data curation, funding acquisition, and methodology.

P Finetti: resources, data curation, formal analysis, supervision, methodology, and writing—original draft.

A Dutour: resources, formal analysis, methodology, and writing—original draft.

J-Y Blay: resources, data curation, funding acquisition, and methodology.

F Bertucci: resources, data curation, formal analysis, supervision, methodology, and writing—original draft.

F Galland: supervision and methodology.

P Naquet: conceptualization, supervision, funding acquisition, methodology, project administration, and writing—original draft, review, and editing.

### Conflict of Interest Statement

The authors declare that they have no conflict of interest.

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
