## [Reviewer comments · Life Science Alliance]

Life Science Alliance

An OMA1 redox site controls mitochondrial homeostasis, sarcoma growth and immunogenicity

Richard Miallot, Virginie Millet, Yann Groult, Angelika Modelska, Lydie Crescence, Sandrine Roulland, Sandrine Henri, Bernard Malissen, Nicolas Brouilly, Laurence Panicot-Dubois, Renaud Vincentelli, Gerlind Sulzenbacher, Pascal Finetti, Aurelie Dutour, Jean-Yves Blay, François Bertucci, Franck Galland, and Philippe Naquet

DOI: <https://doi.org/10.26508/lsa.202201767>

Corresponding author(s): *Philippe Naquet, Centre d'Immunologie de Marseille-Luminy and Richard Miallot, Aix Marseille University*

Review Timeline:

Submission Date:	2022-10-13
Editorial Decision:	2022-11-13
Revision Received:	2023-02-20
Editorial Decision:	2023-03-20
Revision Received:	2023-03-21
Accepted:	2023-03-24

Scientific Editor: Novella Guidi

Transaction Report:

November 13, 2022

Re: Life Science Alliance manuscript #LSA-2022-01767-T

Prof. Philippe Naquet
Centre d'Immunologie de Marseille Luminy
CIML
INSERM-CNRS-Univ. Méditerranée Case 906 Cedex 9
Marseille, PACA 13288
France

Dear Dr. Naquet,

Thank you for submitting your manuscript entitled "An OMA1 redox site controls mitochondrial homeostasis, sarcoma growth and immunogenicity" to Life Science Alliance. The manuscript was assessed by expert reviewers, whose comments are appended to this letter. We invite you to submit a revised manuscript addressing the Reviewer comments.

Thank you for this interesting contribution to Life Science Alliance. We are looking forward to receiving your revised manuscript.

Sincerely,

B. MANUSCRIPT ORGANIZATION AND FORMATTING:

Reviewer #1 (Comments to the Authors (Required)):

Maillot and colleagues investigate the role of a redox-sensing domain of OMA1 protein in sarcoma growth and immunogenicity. The authors hypothesize that interrupting OMA1 function in cancer cells through mutation of this redox-sensing site might interfere with stress response pathways and enhance cancer cell death. The authors investigate the contribution of the Cysteine 403, conserved from yeast to human, in the regulation of OMA1 activity and its impact on mitochondrial function in sarcoma. Mutation of this Cysteine in Alanine leads to mitochondrial alterations and reduced respiratory capacity. This mutation also prevents tumor development in immunocompetent mice, but not in Nude or dendritic cell-deficient mice. The mechanism relies on CD8+ lymphocytes priming in tumor cells, where OMA1 inactivation increases the anti-tumor immunity. In primary tumors, high expression levels of OMA1 or OPA1 correlate with shorter metastasis-free survival whereas lower OPA1 expression is associated with anti-tumor immune signatures. Overall, the study is well conceived and rigorously performed. Therefore, I do not have major concerns.

However, additional points to be further addressed are listed below:

1. Is the MIC60-MIC19 subcomplex affected by OMA1 C403A mutation? And the formation of the MICOS complex? This issue should be verified in both basal and stress conditions.
2. To evaluate whether OMA1 C403A mutation exhibit signs of bioenergetic deficit the oxygen consumption rates should be evaluated. In this regard, galactose-supplemented medium could help by forcing cells to rely on oxidative metabolism.
3. Impairment of L-OPA1 cleavage should destabilize OMA1 function and render cells resistant to apoptotic stimuli, due to attenuated mitochondrial fission and cytochrome c release. To evaluate whether the OMA1 C403A mutation is important for stress-associated membrane remodeling and initiation of the apoptotic program, Annexin V and cytochrome c release should be evaluated in the OMA1 C403A mutation, in normal condition and after treatment with the apoptosis-inducing drugs (i.e. staurosporine).
4. The authors find that the C403A mutation had no significant impact on tumor growth potential in vivo in nude mice, but not in the immunocompetent mice. Since the C403A mutation prevents stress-induced OMA1 activation and downstream cleavage of target proteins, this key finding should be discussed in more detail in the manuscript.
5. The mitochondrial network and cristae structure in the context of OMA1 C403A mutation should be characterized in more detail.

Reviewer #2 (Comments to the Authors (Required)):

Mitochondrial morphology, activity, and signaling are increasingly recognized as key factors contributing to metabolic plasticity of various cancers. However, little is known about the underlying control mechanisms. In this study Miallot et al. have attempted to elucidate the role of OMA1, a mitochondrial protease that recently emerged as a candidate modulator protein in several cancers, in a murine model of sarcoma. Guided by previously published structure-function data from yeast model, the authors describe two conserved cysteine residues comprising a redox-sensing site and demonstrate that one of these residues, Cys403, is required for OMA1 proteolytic activity. Using a combination of biochemical, in vitro, and in vivo animal model studies, the authors show that impaired proteolytic function of the C403A OMA1 mutant is associated with an impaired processing of OPA1 and DELE1, the two key substrates of OMA1 as well as reduced cellular fitness. They also demonstrate that in contrast to the wild type OMA1 harboring sarcoma tumors, the C403A OMA1-expressing tumor xenografts are unable to expand in immunocompetent C57BL/6 mice due to enhanced cell death. Intriguingly, the authors show this is not the case in mice impaired in antigen presentation and consequently, in antitumor immune responses. Finally, the authors examine available clinical data to show that high OMA1 and OPA1 gene expression levels may correlate with poor prognosis in certain subtypes of sarcoma in humans. Overall, this is a novel and interesting study. However, in its current form the manuscript is rather descriptive and has several significant drawbacks, falling short of addressing as to how exactly OMA1 facilitates sarcoma progression.

Specific points:

1. An additional YME1L-specific processing site in OPA1 (termed S3) has been identified recently (Wang et al., 2021). This

information should be included in the introduction and discussion sections. The authors should also discuss a possibility of OMA1 not always engaging in OPA1 processing, which could be one of the ways to interpret their clinical data mining results.

2. The data obtained with OPA1 peptide reporter are not very convincing, likely due to inherent flaws of said reporter (as discussed in e.g. Alavi 2021 and also briefly acknowledged by the authors). A rather unremarkable effect of zinc chelator treatment underscores this notion.
3. Why is there no appreciable change in the steady state levels of OMA1 in Fig.1D? OMA1 activation would be expected result in its autoprocessing. Does any of OMA1 variants of interest show different autoproteolytic pattern upon prolonged incubation with the uncoupler?
4. Related to the above notion, available data indicate that OMA1 is predominantly regulated at the post-translational level. Therefore, unaltered OMA1 transcript levels may not matter that much in this particular case.
5. In Fig. 1B and C as well as in subsequent figures, the abundance of "L-OMA1" from is largely unchanged following CCCP treatment. Why is that? One would expect to see more of that from upon uncoupling. The levels of S-OMA1 do not appear to change much either. In general, the nature of said forms is incompletely understood and debated and therefore should be interpreted with a caution. Showing a control processing pattern for OMA1 catalytic mutant would be very helpful. Furthermore, statistical analysis of these data needs to be included to make valid conclusions.
6. Fig.2E. Multiple reports have established that OMA1-deficient cells are resistant to apoptotic stimuli. If the C403A variant is indeed a partial loss of function mutant, shouldn't it render the cells resistant to bortezomib?
7. The magnification/quality of the electron micrographs shown in Fig.2I is not sufficient to judge the effect of the C403A mutation on mitochondria contact sites. The authors should either highlight what is being shown or present images at a higher magnification.
8. The DELE1 processing data shown in Fig.3B and C are not very convincing. Again, a proper statistical analysis of these data should be included. Furthermore, the authors should present an evidence that downstream effectors of DELE1 (e.g. eIF2a phosphorylation or integrated stress response transcriptional signatures) are altered in the C403A OMA1 expressing cells. They should also include some discussion as to how impaired DELE1 signaling may be contributing to the observed effects.
9. The clinical sample analysis section is confusing. If it is concluded that changes in survival due to different OMA1 expression levels are rather minor and not statistically significant, what is then the main takeaway there? In that setting shifting gears towards OPA1 expression results makes this section look incongruous with the rest of the manuscript.

Additional issues:

1. As already pointed out above, statistical analyses of the data shown in Fig. 1C, F and Fig. 3C need to be included to support authors' conclusions.
2. The manuscript should be thoroughly proofread and edited for syntax.

Reviewer #3 (Comments to the Authors (Required)):

1. The authors present a very interesting study exploring the impact of OMA1 mutation on mitochondrial dynamics and tumor progression. Despite the widely appreciated importance of OMA1 to mitochondrial dynamics and stress response, structural information remains limited; this motivates the authors' use of molecular models, as well as Bohovych et al. 2019, to examine the impact of mutating cysteine 403 to abrogate the disulfide bridge produced at this residue. The authors present compelling evidence showing that the C403A mutation prevents CCCP-induced cleavage of L-OPA1 isoforms. This finding expands the mechanistic understanding of OMA1, adding new importance to OMA1 cysteine residues in mammalian settings. The authors rightly propose that OMA1's importance is heavily context-dependent: as OMA112 cleavage of OPA1 primes the cell for apoptosis, OMA1 inhibition can be anti-apoptotic. On the other hand, disruption of mitochondrial stress response causes bioenergetic dysfunction. These studies are extended to examine tumor growth in mice, as well as OMA1/OPA1 expression signatures in soft tissue sarcoma. The experiments presented provide an important advance in understanding molecular mechanisms of OMA1 function, as well as its role in tumor progression
2. The data are supportive. One experimental issue is the use of MitoTracker for CCCP mitochondrial morphology experiments in Fig. 2A. While MitoTracker is not Nernstian, as rhodamine or TMRE dyes are, i.e. it goes into the mitochondrial matrix and stays there, MitoTracker does rely on transmembrane potential for uptake. Accordingly, CCCP-treated cells will show much lower, suboptimal staining of mitochondria, with some organelles missed altogether. For better confidence in the morphological status of mitochondria in CCCP-treated control and C403A cells, visualization using immunofluorescence, mCherry-mito transfection, or some method that does not rely on membrane potential should be employed.

3. The reviewer recommends the following textual revisions:

-The abstract should be revised to include a description of the specific OMA1 mutation made.

-Check throughout the text for consistency of abbreviations; for example, OMA1 occasionally appears as "OMA-1".

-A comprehensive check of English language usage would be helpful. Occasional awkward phrases are encountered, for example "...a maintenance of mitochondrial activity remains essential".

We thank the reviewers for the interest of their requests. Several new results were added to the manuscript (Figures 1H, 2F,G, H, J and Extended Views 1H, I, 2B, C, G, H) and are discussed in the result sections. We have addressed all the points that were raised and, when necessary, modulated our conclusion (mostly concerning DELE1 cleavage, see EV3C).

A detailed answer to all the requests is provided infra. The revised manuscript is added showing all the corrections and additions in red in the text.

Reviewer #1 (Comments to the Authors (Required)):

1. Is the MIC60-MIC19 subcomplex affected by OMA1 C403A mutation? And the formation of the MICOS complex? This issue should be verified in both basal and stress conditions.

This experiment has been performed and the results shown in Fig 1H (and EV1I) and discussed on page 7. Basically, the mutation does not prevent the formation of the MIB and MICOS complexes detected by BN-PAGE but affects OPA1 cleavage in the MICOS complex.

2. To evaluate whether OMA1 C403A mutation exhibit signs of bioenergetic deficit the oxygen consumption rates should be evaluated. In this regard, galactose-supplemented medium could help by forcing cells to rely on oxidative metabolism.

The results of this experiment are shown in EV2C and discussed on page 7 : both OCR and ECAR were measured. Our results are consistent with the fact that mutant cells, while able to respire, tend to enhance their glycolytic potential to sustain ATP production. We did not need to use galactose-supplemented medium to force cell respiration.

3. Impairment of L-OPA1 cleavage should destabilize OMA1 function and render cells resistant to apoptotic stimuli, due to attenuated mitochondrial fission and cytochrome c release. To evaluate whether the OMA1 C403A mutation is important for stress-associated membrane remodeling and initiation of the apoptotic program, Annexin V and cytochrome c release should be evaluated in the OMA1 C403A mutation, in normal condition and after treatment with the apoptosis-inducing drugs (i.e. staurosporine).

The results of this experiment are shown in Fig 2F and G using two techniques of detection of apoptosis and staurosporine as inducer and discussed on page 8. As observed with OMA1 deficient cells, mutant cells are more resistant to apoptosis.

4. The authors find that the C403A mutation had no significant impact on tumor growth potential in vivo in nude mice, but not in the immunocompetent mice. Since the C403A mutation prevents stress-

induced OMA1 activation and downstream cleavage of target proteins, this key finding should be discussed in more detail in the manuscript.

We discussed the consequences of OPA1 cleavage by OMA1 and the different functions of S-OPA1 isoforms on the organization of cristae and mtDNA release that could be one of the main trigger of immunogenic cues in our model. Elements of this discussion appear on page 3 of the introduction where references were introduced and in the discussion on pages 12-13.

5. The mitochondrial network and cristae structure in the context of OMA1 C403A mutation should be characterized in more detail.

We prepared thinner EM sections and performed a more precise analysis of cristae. New EM pictures with better resolution are shown in Fig 2H, I and J and EV2H as discussed on page 8. The main result shows a modification of cristae area per cell in mutant cells.

Reviewer #2 (Comments to the Authors (Required)):

1. An additional YME1L-specific processing site in OPA1 (termed S3) has been identified recently (Wang et al., 2021). This information should be included in the introduction and discussion sections. The authors should also discuss a possibility of OMA1 not always engaging in OPA1 processing, which could be one of the ways to interpret their clinical data mining results.

Thank you for raising this interesting finding. Specific references are now introduced in the introduction (page 3) and in the discussion (pages 12 and 13) focusing on the complex functions of OPA1 isoforms on mitochondrial fitness.

2. The data obtained with the OPA1 peptide reporter are not very convincing, likely due to inherent flaws of said reporter (as discussed in e.g. Alavi 2021 and also briefly acknowledged by the authors). A rather unremarkable effect of zinc chelator treatment underscores this notion.

We fully agree with this point and added corrective statements in the text on pages 5 and 6. We left the results in EV as it might be relevant for other studies.

3. Why is there no appreciable change in the steady state levels of OMA1 in Fig.1D? OMA1 activation would be expected result in its autoprocessing. Does any of OMA1 variants of interest show different autoproteolytic pattern upon prolonged incubation with the uncoupler?

We performed a kinetic experiment on OMA1 processing following exposure to CCCP comparing CTRL and C403A mutant cells. The results are shown in EV1H. Smaller OMA1 isoforms progressively appear and decline in CTRL cells after 2hrs of treatment. In C403A cells, the total amount of OMA1 is reduced (as previously described) and progressively declines but we could not detect the smallest isoforms of OMA1 in mutant cells that were seen in CTRL cells. Either the autoproteolysis is not complete, or the amount of shorter OMA1 isoforms becomes too low to be detected by western blot analysis. We added these results and discussed their consequences on pages 6 and 12.

4. Related to the above notion, available data indicate that OMA1 is predominantly regulated at the post-translational level. Therefore, unaltered OMA1 transcript levels may not matter that much in this particular case.

We agree with the comment but left the result as a quality control of the edition process in the EV section EV1G.

5. In Fig. 1B and C as well as in subsequent figures, the abundance of "L-OMA1" from is largely unchanged following CCCP treatment. Why is that? One would expect to see more of that from upon uncoupling. The levels of S-OMA1 do not appear to change much either. In general, the nature of said forms is incompletely understood and debated and therefore should be interpreted with a caution. Showing a control processing pattern for OMA1 catalytic mutant would be very helpful. Furthermore, statistical analysis of these data needs to be included to make valid conclusions.

Our answer to this point is connected to Point 3. In our initial experiments, the CCCP treatment was maintained for one hour. As shown in the kinetic experiment on OMA1 processing following exposure to CCCP, the reduction in S-OMA1 becomes detectable only at one hour and more drastically at 2 and 3 hours post CCCP. In Fig 1B and 1D, the amount of S-OMA1 also declined after CCCP treatment. To make it clearer, we presented the results after normalization of band intensities using actin or TOMM20 blotting intensities for total cell or mitochondrial extracts, respectively. The quantification and statistical relevance of these results are shown in Fig1C. We also added statistics on figure 3C representing OMA1 isoforms detected in tumor cells.

6. Fig.2E. Multiple reports have established that OMA1-deficient cells are resistant to apoptotic stimuli. If the C403A variant is indeed a partial loss of function mutant, shouldn't it render the cells resistant to bortezomib?

The results of this experiment are shown in Fig 2F and G (and EV2G) using two techniques of detection of apoptosis and two inducers (staurosporine and bortezomib) and discussed on page 8. As observed with OMA1 deficient cells, mutant cells are more resistant to apoptosis but overall show increased signs of cell death (mostly through necrosis) after bortezomib exposure.

7. The magnification/quality of the electron micrographs shown in Fig.2I is not sufficient to judge the effect of the C403A mutation on mitochondria contact sites. The authors should either highlight what is being shown or present images at a higher magnification.

We prepared thinner EM sections and performed a more precise analysis of cristae. New EM pictures with better resolution are shown in Fig 2H, I and J and EV2H an discussed on page 8. The main result shows a modification of cristae area per cell in C403 mutant tumors.

8. The DELE1 processing data shown in Fig.3B and C are not very convincing. Again, a proper statistical analysis of these data should be included. Furthermore, the authors should present an evidence that downstream effectors of DELE1 (e.g. eIF2a phosphorylation or integrated stress response transcriptional signatures) are altered in the C403A OMA1 expressing cells. They should also include

some discussion as to how impaired DELE1 signaling may be contributing to the observed effects.

We agree with this comment and performed new experiments on cultured cells to try to clarify our conclusions. The initial western blot analysis was performed on 4 independent tumor cell populations extracted from the tumor masses of each genotype. As shown in Fig EV3B, our results are at the limit of the statistical significance. Although we could have added more tumor samples, we think that the results might not be dramatically different.

To tackle this issue using a different approach, we treated cultured MCA cells with oligomycin to induce DELE1 cleavage in vitro. The results shown in FigEV3C showed that DELE-1 already exists as a mix of L- and S-DELE1 before adding oligomycin. This suggests that in these tumor cells, there might be a background level of ER stress possibly linked to the fact that mitochondrial activity is altered at steady state and furthermore, signs of altered mitochondria/ER contacts could be detected by EM analysis. We performed several additional experiments to detect evidence of differential ER stress (eIF2 α phosphorylation, ATF4 blotting, analysis by qRT-PCR of ATF4 target gene expression) between CTRL and C403A cells but the results were not sufficiently discriminative to be shown although some minor differences could be detected. We discussed the limits of our analysis and preferred to retrieve the statement that DELE1 processing is affected by OMA1 mutation (see text on page 9). We had also the possibility of excluding these results from the manuscript but finally thought that even if not definitively conclusive, they might be of interest for the research community. We are ready to reconsider this issue naturally.

9. The clinical sample analysis section is confusing. If it is concluded that changes in survival due to different OMA1 expression levels are rather minor and not statistically significant, what is then the main takeaway there? In that setting shifting gears towards OPA1 expression results makes this section look incongruous with the rest of the manuscript.

One of the difficulty encountered in sarcoma research is due to the high level of heterogeneity combined with the scarcity of tumor subtypes. Given the heterogeneity of the published results showing positive or negative correlation of OMA1 expression with the prognosis of tumors (mostly carcinomas), we think that it is useful to provide information on sarcomas. Our study relies on a large collection of samples from different origins. Describing variations in OMA1 and OPA1 expression in different tumor types is relevant. As seen in Fig 5B, the correlation between OMA1 expression and 5y MFS does not reach statistical significance. In contrast, results for OPA1 are clear and furthermore highly connected to the detection of immune signatures. Since OPA1 processing is one of the main consequence of OMA1 activation in stressed tumor cells, one can think that over or underexpression of OPA1 transcripts may be the consequence of a selective pressure among tumor clones reflecting adaptation of the tumor to mitochondrial stress. Therefore, we modulated the amount of information shown in the result section (page 11) and tried to better discuss our results on page 13.

Additional issues:

1. As already pointed out above, statistical analyses of the data shown in Fig. 1C, F and Fig. 3C need to be included to support authors' conclusions.

This has been done.

2. The manuscript should be thoroughly proofread and edited for syntax.

The manuscript has been read by native english-speaking researchers. We hope to have corrected syntax errors.

Reviewer #3 (Comments to the Authors (Required)):

2. The data are supportive. One experimental issue is the use of MitoTracker for CCCP mitochondrial morphology experiments in Fig. 2A. While MitoTracker is not Nerstian, as rhodamine or TMRE dyes are, i.e. it goes into the mitochondrial matrix and stays there, MitoTracker does rely on transmembrane potential for uptake. Accordingly, CCCP-treated cells will show much lower, suboptimal staining of mitochondria, with some organelles missed altogether. For better confidence in the morphological status of mitochondria in CCCP-treated control and C403A cells, visualization using immunofluorescence, mCherry-mito transfection, or some method that does not rely on membrane potential should be employed.

This comment is perfectly valid. One could argue that despite the partial reduction in staining intensity, the morphology of mitochondria is still detectable in CCCP treated cells. However, to comfort our results, we repeated the experiment using TOMM20 immunostaining as an alternative. The results are shown in Fig EV2B and show the same phenotype in C403A mutant cells. We also obtained similar results using MitoTracker Green known to be a relevant marker of mitochondrial mass. Again the same phenotype was detected as shown infra (2 independent images per condition).

3. The reviewer recommends the following textual revisions:
-The abstract should be revised to include a description of the specific OMA1 mutation made.

Done

-Check throughout the text for consistency of abbreviations; for example, OMA1 occasionally appears as "OMA-1".

Done.

-A comprehensive check of English language usage would be helpful. Occasional awkward phrases are encountered, for example ...a maintenance of mitochondrial activity remains essential".

The manuscript has been read by native english-speaking researchers. We hope to have corrected the remaining syntax errors.

March 20, 2023

RE: Life Science Alliance Manuscript #LSA-2022-01767-TR

Prof. Philippe Naquet
Centre d'Immunologie de Marseille-Luminy
CIML
INSERM-CNRS-Univ. Méditerranée Case 906 Cedex 9
Marseille, PACA 13288
France

Dear Dr. Naquet,

Thank you for submitting your revised manuscript entitled "An OMA1 redox site controls mitochondrial homeostasis, sarcoma growth and immunogenicity". We would be happy to publish your paper in Life Science Alliance pending final revisions necessary to meet our formatting guidelines.

- please address Reviewer 2's remaining point #4 and additional issues
- please rename your EV figures as supplementary figures and update the figure callouts in the manuscript text accordingly
- please relabel your EV4 and EV 5 figures as tables and label these tables accordingly; your table files must be uploaded as editable doc or excel files
- your 2 Supplemental Material files can be uploaded as Supplemental Tables and labeled as such

Figure Check:

- please double-check your figure legend for Figure EV3 and make sure your panels are labeled correctly
- please add panel D for Figure 5 to the figure legend and add panel E to your Figure Ev3 legend
- Figure 2H scale bars are not easy to read

A. FINAL FILES:

B. MANUSCRIPT ORGANIZATION AND FORMATTING:

Sincerely,

Reviewer #1 (Comments to the Authors (Required)):

The authors have satisfactorily addressed my previous concerns and the manuscript is now acceptable for publication.

Reviewer #2 (Comments to the Authors (Required)):

The revised manuscript has been generally improved with additional new data and statistical analyses. The discussion has also been improved in length and focus. However, there are several remaining points that should be addressed to merit the acceptance.

1. The reviewer suggests removing the data obtained with OPA1 peptide reporter as they are not making a compelling point and detract from the study's value and manuscript's readability.
2. Fig.1B. This experiment needs to be properly controlled with a bona fide catalytic mutant variant of OMA1. As presented, the data do not provide sound evidence for C403A mutant being catalytically impaired.
3. Fig.3B. The reviewer suggests these data to be removed or presented as supplemental information as they don't seem to flow well with the rest of the manuscript. If the authors suspect DELE1 to be a significant contributor, this aspect of the study should be addressed in more detail.
4. Several key graphs (e.g. Fig.4D and Fig.4G) are lacking indications of significance levels. Related to this notion, for all figures with error bars - please clarify in the figure legend what the error bars are: Standard deviation, or standard error of the mean?

Additional issues:

The reviewer is pleased to hear that the manuscript has been examined by native English speakers, however occasional typos and syntax errors still can be spotted in the text. Here are some examples:

p.2, the 4th and the last sentences of the Abstract section.

p.3, the 7th sentence in the second paragraph - the correct term is "hyperfusion".

p.12, the last sentence in the Results section is quite awkward.

Reviewer #3 (Comments to the Authors (Required)):

The authors have addressed my concerns and the manuscript is suitable for publication.

March 24, 2023

RE: Life Science Alliance Manuscript #LSA-2022-01767-TRR

Prof. Philippe Naquet
Centre d'Immunologie de Marseille-Luminy
CIML
INSERM-CNRS-Univ. Méditerranée Case 906 Cedex 9
Marseille, PACA 13288
France

Dear Dr. Naquet,

Thank you for submitting your Research Article entitled "An OMA1 redox site controls mitochondrial homeostasis, sarcoma growth and immunogenicity". It is a pleasure to let you know that your manuscript is now accepted for publication in Life Science Alliance. Congratulations on this interesting work.

DISTRIBUTION OF MATERIALS:

Again, congratulations on a very nice paper. I hope you found the review process to be constructive and are pleased with how the manuscript was handled editorially. We look forward to future exciting submissions from your lab.

Sincerely,
